# Robust Fair Clustering: A Novel Fairness Attack and Defense Framework

**Anshuman Chhabra[†], Peizhao Li[§], Prasant Mohapatra[†], and Hongfu Liu[§]**
[†] Department of Computer Science, University of California, Davis
{chhabra,pmohapatra}@ucdavis.edu
[§] Michtom School of Computer Science, Brandeis University
{peizhaoli,hongfuliu}@brandeis.edu

## Abstract

Clustering algorithms are widely used in many societal resource allocation applications, such as loan approvals and candidate recruitment, among others, and hence, biased or unfair model outputs can adversely impact individuals that rely on these applications. To this end, many *fair* clustering approaches have been recently proposed to counteract this issue. Due to the potential for significant harm, it is essential to ensure that fair clustering algorithms provide consistently fair outputs even under adversarial influence. However, fair clustering algorithms have not been studied from an adversarial attack perspective. In contrast to previous research, we seek to bridge this gap and conduct a robustness analysis against fair clustering by proposing a novel *black-box fairness attack*. Through comprehensive experiments[1], we find that state-of-the-art models are highly susceptible to our attack as it can reduce their fairness performance significantly. Finally, we propose Consensus Fair Clustering (CFC), the first *robust fair clustering* approach that transforms consensus clustering into a fair graph partitioning problem, and iteratively learns to generate fair cluster outputs. Experimentally, we observe that CFC is highly robust to the proposed attack and is thus a truly robust fair clustering alternative.

## 1 Introduction

Machine learning models are ubiquitously utilized in many applications, including high-stakes domains such as loan disbursement (Tsai & Chen, 2010), recidivism prediction (Berk et al., 2021; Ferguson, 2014), hiring and recruitment (Roy et al., 2020; Pombo, 2019), among others. For this reason, it is of paramount importance to ensure that decisions derived from such predictive models are unbiased and fair for all individuals treated (Mehrabi et al., 2021a). In particular, this is the main motivation behind *group-level fair* learning approaches (Celis et al., 2021a; Li & Liu, 2022; Song et al., 2021), where the goal is to generate predictions that do not *disparately impact* individuals from minority protected groups (such as ethnicity, sex, etc.). It is also worthwhile to note that this problem is technically challenging because there exists an inherent fairness-performance tradeoff (Dutta et al., 2020), and thus fairness needs to be improved while ensuring approximate preservation of model predictive performance. This line of research is even more pertinent for data clustering, where error rates cannot be directly assessed using class labels to measure disparate impact. Thus, many approaches have been recently proposed to make clustering models group-level fair (Chierichetti et al., 2017; Backurs et al., 2019; Kleindessner et al., 2019a; Chhabra et al., 2022b). In a nutshell, these approaches seek to improve fairness of clustering outputs with respect to some fairness metrics, which ensure that each cluster contains approximately the same proportion of samples from each protected group as they appear in the dataset.

While many fair clustering approaches have been proposed, it is of the utmost importance to ensure that these models provide fair outputs even in the presence of an adversary seeking to degrade fairness utility. Although there are some pioneering attempts on fairness attacks against supervised learning models (Solans et al., 2020; Mehrabi et al., 2021b), unfortunately, none of these works propose defense approaches. Moreover, in the unsupervised scenario, fair clustering algorithms have not yet

---

been explored from an adversarial attack perspective, which leaves the whole area of unsupervised fair clustering in potential danger. This leads us to our fundamental research questions in this paper:

*Are fair clustering algorithms vulnerable to adversarial attacks that seek to decrease fairness utility, and if such attacks exist, how do we develop an adversarially robust fair clustering model?*

**Contributions**. In this paper, we answer both these questions in the affirmative by making the following contributions:

- We propose a novel *black-box* adversarial attack against clustering models where the attacker can perturb a small percentage of protected group memberships and yet is able to degrade the fairness performance of state-of-the-art fair clustering models significantly (Section 2). We also discuss how our attack is critically different from existing adversarial attacks against clustering performance and why they cannot be used for the proposed threat model.

- Through extensive experiments using our attack approach, we find that existing fair clustering algorithms are not robust to adversarial influence, and are extremely volatile with regards to fairness utility (Section 2.2). We conduct this analysis on a number of real-world datasets, and for a variety of clustering performance and fairness utility metrics.

- To achieve truly robust fair clustering, we propose the *Consensus Fair Clustering* (CFC) model (Section 3) which is highly resilient to the proposed fairness attack. To the best of our knowledge, CFC is the first defense approach for fairness attacks, which makes it an important contribution to the unsupervised ML community.

**Preliminaries and Notation**. Given a tabular dataset $X=\{x_i\}\in\mathbb{R}^{n\times d}$ with $n$ samples and $d$ features, each sample $x_i$ is associated with a protected group membership $g(x_i)\in[L]$, where $L$ is the total number of protected groups, and we denote group memberships for the entire dataset as $G=\{g(x_i)\}_{i=1}^{n}\in\mathbb{N}^n$. We also have $H=\{H_1, H_2, ..., H_L\}$ and $H_l$ is the set of samples that belong to $l$-th protected group. A clustering algorithm $\mathcal{C}(X,K)$ takes as input the dataset $X$ and a parameter $K$, and outputs labeling where each sample belongs to one of $K$ clusters (Xu & Wunsch, 2005). That is, each point is clustered in one of the sets $\{C_1, C_2, ..., C_K\}$ with $\cup_{k=1}^{K}C_k = X$. Based on the above, a *group-level fair* clustering algorithm $\mathcal{F}(X,K,G)$ (Chierichetti et al., 2017) can be defined similarly to $\mathcal{C}$, where $\mathcal{F}$ takes as input the protected group membership $G$ along with $X$ and $K$, and outputs fair labeling that is expected to be more *fair* than the clustering obtained via the original unfair/vanilla clustering algorithm with respect to a given fairness utility function $\phi$. That is, $\phi(\mathcal{F}(X,K,G),G) \leq \phi(\mathcal{C}(X,K),G)$. Note that $\phi$ can be defined to be any fairness utility metric, such as Balance and Entropy (Chhabra et al., 2021a; Mehrabi et al., 2021a).

## 2 FAIRNESS ATTACK

In this section, we study the attack problem on fair clustering. Specifically, we propose a novel attack that aims to reduce the fairness utility of fair clustering algorithms, as opposed to traditional adversarial attacks that seek to decrease clustering performance Cinà et al. (2022). To our best knowledge, although there are a few pioneering attempts toward fairness attack (Mehrabi et al., 2021b; Solans et al., 2020), all of them consider the supervised setting. Our proposed attack exposes a novel problem prevalent with fair clustering approaches that has not been given considerable attention yet– as the protected group memberships are input to the fair clustering optimization problem, they can be used to disrupt the fairness utility. We study attacks under the *black-box* setting, where the attacker has no knowledge of the fair clustering algorithm being used. Before formulating the problem in detail, we first define the threat model as the adversary and then elaborate on our proposed attack.

### 2.1 THREAT MODEL TO ATTACK FAIRNESS

**Threat Model**. Take the customer segmentation (Liu & Zhou, 2017; Nazari & Sheikholeslami, 2021) as an example and assume that the sensitive attribute considered is *age* with 3 protected groups: {*youth, adult, senior*}. Now, we can motivate our threat model as follows: the adversary can control a small portion of individuals' protected group memberships (either through social engineering, exploiting a security flaw in the system, etc.); by changing their protected group memberships, the adversary aims to disrupt the fairness utility of the fair algorithm on other uncontrolled groups. That is, there would be an overwhelming majority of some protected group samples over others in clusters.

This would adversely affect the youth and senior groups, as they are more vulnerable and less capable of enforcing self-prevention. The attacker could carry out this attack for profit or anarchistic reasons.

Our adversary has partial knowledge of the dataset $X$ but not the fair clustering algorithm $\mathcal{F}$. However, they can query $\mathcal{F}$ and observe cluster outputs. This assumption has been used in previous adversarial attack research against clustering (Cinà et al., 2022; Chhabra et al., 2020a; Biggio et al., 2013)). They can access and switch/change the protected group memberships for a small subset of samples in $G$, denoted as $G_A \subseteq G$. Our goal of the fairness attack is to change the protected group memberships of samples in $G_A$ such that the fairness utility value decreases for the remaining samples in $G_D = G \backslash G_A$. As clustering algorithms (Von Luxburg, 2007) and their fair variants (Kleindessner et al., 2019b) are trained on the input data to generate labeling, this attack is a *training-time* attack. Our attack can also be motivated by considering that fair clustering outputs change with any changes made to protected group memberships $G$ or the input dataset $X$. We can formally define the fairness attack as follows:

**Definition 2.1** (Fairness Attack). *Given a fair clustering algorithm $\mathcal{F}$ that can be queried for cluster outputs, dataset $X$, samples' protected groups $G$, and $G_A \subseteq G$ is a small portion of protected groups that an adversary can control, the fairness attack is that the adversary aims to reduce the fairness of clusters outputted via $\mathcal{F}$ for samples in $G_D = G \setminus G_A \subseteq G$ by perturbing $G_A$.*

**Attack Optimization Problem**. Based on the above threat model, the attack optimization problem can be defined analytically. For ease of notation, we define two mapping functions:

- $\eta$ : Takes $G_A$ and $G_D$ as inputs and gives output $G = \eta(G_A, G_D)$ which is the combined group memberships for the entire dataset. Note that $G_A$ and $G_D$ are interspersed in the entire dataset in an unordered fashion, which motivates the need for this mapping.

- $\theta$ : Takes $G_D$ and an output cluster labeling from a clustering algorithm for the entire dataset as input, returns the cluster labels for only the subset of samples that have group memberships in $G_D$. That is, if the clustering output is $\mathcal{C}(X, K)$, we can obtain cluster labels for samples in $G_D$ as $\theta(\mathcal{C}(X, K), G_D)$.

Based on the above notations, we have the following optimization problem for the attacker:

$$\min_{G_A} \quad \phi(\theta(O, G_D), G_D) \ \text{ s.t. } \ O = \mathcal{F}(X, K, \eta(G_A, G_D)). \tag{1}$$

The above problem is a two-level hierarchical optimization problem (Anandalingam & Friesz, 1992) with optimization variable $G_A$, where the lower-level problem is the fair clustering problem $\mathcal{F}(X, K, \eta(G_A, G_D))$, and the upper-level problem aims to reduce the fairness utility $\phi$ of the clustering obtained on the set of samples in $G_D$. Due to the black-box nature of our attack, both the upper- and lower-level problems are highly non-convex and closed-form solutions to the hierarchical optimization cannot be obtained. In particular, hierarchical optimization even with linear upper- and lower-level problems has been shown to be NP-Hard (Ben-Ayed & Blair, 1990), indicating that such problems cannot be solved by exact algorithms. We will thus resort to generally well-performing heuristic algorithms for obtaining solutions to the problem in Eq. (1).

**Solving the Attack Problem**. The aforementioned attack problem in Eq. (1) is a non-trivial optimization problem, where the adversary has to optimize $G_A$ such that overall clustering fairness for the remaining samples in $G_D$ decreases. Since $\mathcal{F}$ is a black-box and unknown to the attacker, first- or second-order approaches (such as gradient descent) cannot be used to solve the problem. Instead, we utilize zeroth-order optimization algorithms to solve the attack problem. In particular, we use RACOS (Yu et al., 2016) due to its known theoretical guarantees on discrete optimization problems. Moreover, our problem belongs to the same class as protected group memberships are discrete labels.

**Discussion**. Note that Chhabra et al. (2021b) propose a theoretically motivated fairness disrupting attack for k-median clustering; however, this cannot be utilized to tackle our research problem for the following reasons: (1) their attack only works for k-median vanilla clustering, thus not constituting a black-box attack on fair algorithms, (2) their attack aims to poison a subset of the input data and not the protected group memberships thus leading to a more common threat model different from us. We also cannot use existing adversarial attacks against clustering algorithms (Cinà et al., 2022; Chhabra et al., 2020a) as they aim to reduce clustering performance and do not optimize for a reduction in fairness utility. Thus, these attacks might not always lead to a reduction in fairness utility. Hence, these threat models are considerably different from our case.

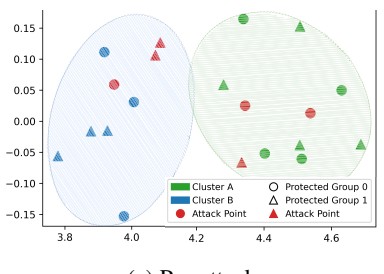 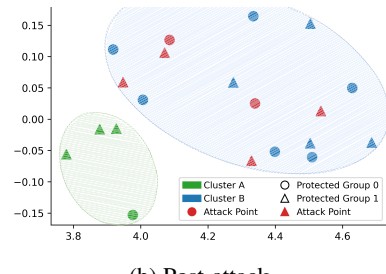

|           |           |
|:---------:|:---------:|
| (a) Pre-attack | (b) Post-attack |

Figure 1: Pre-attack and post-attack clusters of the SFD fair clustering algorithm on the synthetic toy data. The labels of Cluster A and Cluster B are shown in green and blue, and these samples in two clusters belong to $G_D$. The ○ and △ markers represent the two protected groups, and points in red are the attack points that belong to $G_A$. Observe that before the attack, the SFD algorithm obtains a perfect Balance of 1.0. However, after the attack, once the attacker has optimized the protected group memberships for the attack points, the SFD clustering has become less fair with Balance = 0.5.

## 2.2 RESULTS FOR THE ATTACK

**Datasets.** We utilize one synthetic and four real-world datasets in our experiments. The details of our synthetic dataset will be illustrated in the below paragraph of *Performance on the Toy Dataset*. *MNIST-USPS*: Similar to previous work in deep fair clustering (Li et al., 2020), we construct *MNIST-USPS* dataset using all the training digital samples from MNIST (LeCun, 1998) and USPS dataset (LeCun, 1990), and set the sample source as the protected attribute (MNIST/USPS). *Office-31*: The *Office-31* dataset (Saenko et al., 2010) was originally used for domain adaptation and contains images from 31 different categories with three distinct source domains: Amazon, Webcam, and DSLR. Each domain contains all the categories but with different shooting angles, lighting conditions, etc. We use DSLR and Webcam for our experiments and let the domain source be the protected attribute for this dataset. Note that we also conduct experiments on the *Inverted UCI DIGITS dataset* (Xu et al., 1992) and *Extended Yale Face B* dataset (Lee et al., 2005), and provide results for these in the appendix.

**Fair Clustering Algorithms.** We include three state-of-the-art fair clustering algorithms: Fair K-Center (KFC) (Harb & Lam, 2020), Fair Spectral Clustering (FSC) (Kleindessner et al., 2019b) and Scalable Fairlet Decomposition (SFD) (Backurs et al., 2019) for fairness attack, and these methods employ different traditional clustering algorithms on the backend: KFC uses k-center, SFD uses k-median, and FSC uses spectral clustering. Implementation details and hyperparameter values for these algorithms are provided in Appendix A.

**Protocol.** Fair clustering algorithms, much like their traditional counterparts, are extremely sensitive to initialization (Celebi et al., 2013). Differences in the chosen random seed can lead to widely different fair clustering outputs. Thus, we use 10 different random seeds when running the SFD, FSC, and KFC fair algorithms and obtain results. We also uniformly randomly sampled $G_A$ and $G_D$ initially to select these sets such that the fairness utility (i.e. Balance) before the attack is a reasonably high enough value to attack. The size of $G_A$ is varied from 0% to 30% to see how this affects the attack trends. Furthermore, for the zeroth-order attack optimization, we always attack the Balance metric (unless the fair algorithm always achieves 0 Balance in which case we attack Entropy). Note that Balance is a harsher fairness metric than Entropy and hence should lead to a more successful attack. As a performance baseline, we also compare with a random attack where instead of optimizing $G_A$ to reduce fairness utility on $G_D$, we uniformly randomly pick group memberships in $G_A$.

**Evaluation Metrics.** We use four metrics along two dimensions: fairness utility and clustering utility for performance evaluation. For clustering utility, we consider Unsupervised Accuracy (ACC) (Li & Ding, 2006), and Normalized Mutual Information (NMI) (Strehl & Ghosh, 2002). For fairness utility we consider Balance (Chierichetti et al., 2017) and Entropy (Li et al., 2020). The definitions for these metrics are provided in Appendix B. These four metrics are commonly used in the fair clustering literature. Note that for each of these metrics, the higher the value obtained, the better the utility. For fairness, Balance is a better metric to attack, because a value of 0 means that there is a cluster that has 0 samples from one or more protected groups. Finally, the attacker does not care about the clustering utility as long as changes in utility do not reveal that an attack has occurred.

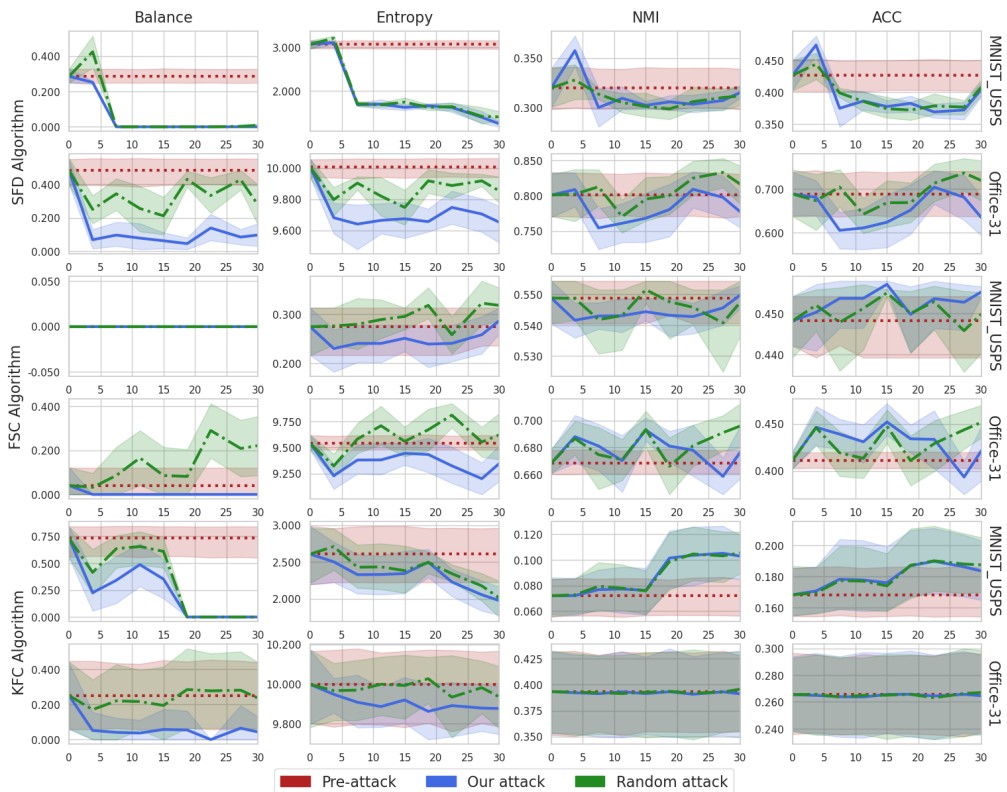

Figure 2: Attack results for *MNIST-USPS* & *Office-31* (x-axis: % of samples attacker can poison).

**Performance on the Toy Dataset**. To demonstrate the effectiveness of the poisoning attack, we also generate a 2-dimensional 20-sample synthetic toy dataset using an isotropic Gaussian distribution, with standard deviation = 0.12, and centers located at (4,0) and (4.5, 0). Out of these 20 points, we designate 14 to belong to $G_D$ and the remaining 6 to belong to $G_A$. The number of clusters is set to $k = 2$. These are visualized in Figure 1. We generate cluster outputs using the SFD fair clustering algorithm before the attack (Figure 1a), and after the attack (Figure 1b). Before the attack, SFD achieves perfect fairness with a Balance of $1.0$ as for each protected group and both Cluster A and Cluster B, we have Balance $\frac{4/8}{7/14} = 1.0$ and $\frac{3/6}{7/14} = 1.0$, respectively. Moreover, performance utility is also high with NMI = 0.695 and ACC = 0.928. However, after the attack, fairness utility decreases significantly. The attacker changes protected group memberships of the attack points, and this leads to the SFD algorithm trying to find a more optimal *global* solution, but in that, it reduces fairness for the points belonging to $G_D$. Balance drops to $0.5$ as for Cluster A and Protected Group 0 we have Balance $\frac{1/4}{7/14} = 0.5$, leading to a $50\%$ decrease. Entropy also drops down to $0.617$ from $0.693$. Performance utility decreases in this case, but is still satisfactory with NMI = 0.397 and ACC = 0.785. Thus, it can be seen that our attack can disrupt the fairness of fair clustering algorithms significantly.

**Performance on Real-World Datasets**. We show the pre-attack and post-attack results *MNIST-USPS* and *Office-31* by our attack and random attack in Figure 2. It can be observed that our fairness attack consistently outperforms the random attack baseline in terms of both fairness metrics: Balance and Entropy. Further, our attack always leads to lower fairness metric values than the pre-attack values obtained, while this is often not the case for the random attack, Balance and Entropy increase for the random attack on the FSC algorithm on *Office-31* dataset. Interestingly, even though we do not optimize for this, clustering performance utility (NMI and ACC) do not drop significantly and even increase frequently. For example, NMI/ACC for FSC on *Office-31* (Figure 2, Row 4, Column 3-4) and NMI/ACC for KFC on *MNIST-USPS* (Figure 2, Row 5, Column 3-4). Thus, the attacker can easily subvert the defense as the clustering performance before and after the attack does not decrease drastically and at times even increases. We also conduct the Kolmogorov-Smirnov statistical test

Table 1: Results for our attack and random attack when 15% group membership labels are switched.

| Algorithms | Metrics | MNIST-USPS | | | | |
| --- | --- | --- | --- | --- | --- | --- |
| | | Pre-Attack | Our Attack | Change (%) | Random Attack | Change (%) |
| SFD | Balance | $0.286 \pm 0.065$ | $0.000 \pm 0.000$ | (-)100.0 | $0.000 \pm 0.000$ | (-)100.0 |
| | Entropy | $3.070 \pm 0.155$ | $1.621 \pm 0.108$ | (-)47.12 | $1.743 \pm 0.156$ | (-)43.23 |
| | NMI | $0.320 \pm 0.033$ | $0.302 \pm 0.007$ | (-)5.488 | $0.301 \pm 0.019$ | (-)5.847 |
| | ACC | $0.427 \pm 0.040$ | $0.378 \pm 0.015$ | (-)11.54 | $0.374 \pm 0.026$ | (-)12.35 |
| FSC | Balance | $0.000 \pm 0.000$ | $0.000 \pm 0.000$ | (-)100.0 | $0.000 \pm 0.000$ | (-)100.0 |
| | Entropy | $0.275 \pm 0.077$ | $0.251 \pm 0.041$ | (-)8.576 | $0.295 \pm 0.036$ | (+)7.598 |
| | NMI | $0.549 \pm 0.011$ | $0.544 \pm 0.007$ | (-)0.807 | $0.552 \pm 0.006$ | (+)0.491 |
| | ACC | $0.448 \pm 0.012$ | $0.457 \pm 0.002$ | (+)1.971 | $0.455 \pm 0.002$ | (+)1.510 |
| KFC | Balance | $0.730 \pm 0.250$ | $0.352 \pm 0.307$ | (-)51.87 | $0.608 \pm 0.237$ | (-)16.72 |
| | Entropy | $2.607 \pm 0.607$ | $2.343 \pm 0.444$ | (-)10.12 | $2.383 \pm 0.490$ | (-)8.595 |
| | NMI | $0.072 \pm 0.024$ | $0.076 \pm 0.029$ | (+)5.812 | $0.076 \pm 0.027$ | (+)5.330 |
| | ACC | $0.168 \pm 0.026$ | $0.176 \pm 0.034$ | (+)4.581 | $0.174 \pm 0.031$ | (+)3.419 |

| Algorithms | Metrics | Office-31 | | | | |
| --- | --- | --- | --- | --- | --- | --- |
| | | Pre-Attack | Our Attack | Change (%) | Random Attack | Change (%) |
| SFD | Balance | $0.484 \pm 0.129$ | $0.062 \pm 0.080$ | (-)87.21 | $0.212 \pm 0.188$ | (-)56.34 |
| | Entropy | $10.01 \pm 0.098$ | $9.675 \pm 0.187$ | (-)3.309 | $9.748 \pm 0.181$ | (-)2.581 |
| | NMI | $0.801 \pm 0.050$ | $0.768 \pm 0.058$ | (-)4.110 | $0.795 \pm 0.053$ | (-)0.726 |
| | ACC | $0.688 \pm 0.082$ | $0.624 \pm 0.098$ | (-)9.397 | $0.668 \pm 0.091$ | (-)2.908 |
| FSC | Balance | $0.041 \pm 0.122$ | $0.000 \pm 0.000$ | (-)100.0 | $0.086 \pm 0.172$ | (+)110.8 |
| | Entropy | $9.538 \pm 0.113$ | $9.443 \pm 0.178$ | (-)0.997 | $9.558 \pm 0.226$ | (+)0.207 |
| | NMI | $0.669 \pm 0.014$ | $0.693 \pm 0.014$ | (+)3.659 | $0.693 \pm 0.022$ | (+)3.711 |
| | ACC | $0.411 \pm 0.014$ | $0.452 \pm 0.027$ | (+)9.904 | $0.447 \pm 0.030$ | (+)8.817 |
| KFC | Balance | $0.250 \pm 0.310$ | $0.057 \pm 0.172$ | (-)77.07 | $0.194 \pm 0.301$ | (-)22.55 |
| | Entropy | $9.997 \pm 0.315$ | $9.919 \pm 0.189$ | (-)0.786 | $9.992 \pm 0.250$ | (-)0.051 |
| | NMI | $0.393 \pm 0.064$ | $0.391 \pm 0.063$ | (-)0.483 | $0.393 \pm 0.067$ | (-)0.160 |
| | ACC | $0.265 \pm 0.048$ | $0.266 \pm 0.049$ | (+)0.032 | $0.265 \pm 0.051$ | (-)0.162 |

(Massey Jr, 1951) between our attack and the random attack result distribution for the fairness utility metrics (Balance and Entropy) to see if the mean distributions are significantly different. We find that for the *Office-31* dataset our attack is statistically significant in terms of fairness values and obtains p-values of less than $< 0.01$. For *MNIST-USPS*, the results are also statistically significant except for the cases when the utility reduces quickly to the same value. For example, it can be seen that Balance goes to 0 for SFD on *MNIST-USPS* in Figure 2 (Row 1, Column 1) fairly quickly. For these distributions, it is intuitive why we cannot obtain statistically significant results, as the two attack distributions become identical. We provide detailed test statistics in Appendix C. We also provide the attack performance on *Inverted UCI DIGITS dataset* (Xu et al., 1992) and *Extended Yale Face B* dataset (Lee et al., 2005) in Appendix E.

Furthermore, to better compare our attack with the random attack, we present the results when $15\%$ group memberships are switched in Table 1. As can be seen, for all fair clustering algorithms and datasets, our attack leads to a more significant reduction in fairness utility for both the *MNIST-USPS* and *Office-31* datasets. In fact, as mentioned before, the random attack at times leads to an *increase* in fairness utility compared to before the attack (refer to FSC Balance/Entropy on *Office-31*). In contrast, our attack always leads to a reduction in fairness performance. For example, for the KFC algorithm and *Office-31* dataset, our attack achieves a reduction in Balance of $77.07\%$ whereas the random attack reduces Balance by only $22.52\%$. However, it is important to note here that existing fair clustering algorithms are very *volatile* in terms of performance, as the random attack can also lead to fairness performance drops, especially for the SFD algorithm (refer to Figure 2 for visual analysis). This further motivates the need for a more robust fair clustering algorithm.

## 3 FAIRNESS DEFENSE: CONSENSUS FAIR CLUSTERING

Beyond proposing a fairness attack, we also provide our defense against the proposed attack. According to the fairness attack in Definition 2.1, we also provide a definition of *robust fair clustering* to defend against the fairness attack, followed by our proposed defense algorithm.

**Definition 3.1** (Robust Fair Clustering). *Given the dataset $X$, samples' protected groups $G$, and $G_A \subseteq G$ is a small portion of protected groups that an adversary can control, a fair clustering*

*algorithm $\mathcal{F}$ is considered to be robust to the fairness attack if the change in fairness utility on $G_D = G \setminus G_A \subseteq G$ before the attack and after the attack is by a marginal amount, or if the fairness utility on $G_D$ increases after the attack.*

### 3.1 DEFENSE ALGORITHM TO FAIRNESS ATTACK

To achieve a robust fair clustering, our defense utilizes consensus clustering (Liu et al., 2019; Fred & Jain, 2005; Lourenço et al., 2015; Fern & Brodley, 2004) combined with fairness constraints to ensure that cluster outputs are robust to the above attack. Consensus clustering has been widely renowned for its robustness and consistency properties but to the best of our knowledge, no other work has utilized consensus clustering concepts in the fair clustering scenario. Specifically, we propose Con-

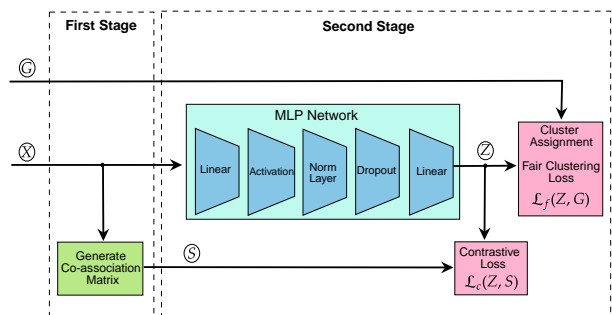

Figure 3: Our proposed CFC framework.

sensus Fair Clustering (CFC) shown in Figure 3, which first transforms the consensus clustering problem into a graph partitioning problem, and then utilizes a novel graph-based neural network architecture to learn representations for fair clustering. CFC has two stages to tackle the attack challenge at the data and algorithm level. The first stage is to sample a subset of training data and run cluster analysis to obtain the basic partition and the co-association matrix. Since poisoned samples are a tiny portion of the whole training data, the probability of these being selected into the subset is also small, which decreases their negative impact. In the second stage, CFC fuses the basic partitions with a fairness constraint and further enhances the algorithmic robustness.

**First Stage: Generating Co-Association Matrix**. In this first stage of CFC, we will generate $r$ basic partitions $\Pi = \{\pi_1, \pi_2, ..., \pi_r\}$. For each basic partition $\pi_i$, we first get a sub dataset $X_i$ by random sample/feature selection and run K-means (Lloyd, 1982) to obtain a basic partition $\pi_i$. Such a process is repeated $r$ times such that $\cup_{i=1}^r X_i = X$. Given that $u$ and $v$ are two samples and $\pi_i(u)$ is the label category of $u$ in basic partition $\pi_i$, and following the procedure of consensus clustering, we summarize the basic partitions into a co-association matrix $S \in \mathbb{R}^{n \times n}$ as $S_{uv} = \sum_{i=1}^r \delta(\pi_i(u), \pi_i(v))$, where $\delta(a, b) = 1$ if $a = b$; otherwise, $\delta(a, b) = 0$. The co-association matrix not only summarizes the categorical information of basic partitions into a pair-wise relationship, but also provides an opportunity to transform consensus clustering into a graph partitioning problem, where we can learn a fair graph embedding that is resilient to the protected group membership poisoning attack.

**Second Stage: Learning Graph Embeddings for Fair Clustering**. In the second stage of CFC, we aim to find an optimal consensus and fair partition based on the feature matrix $X$, basic partitions $\Pi$, and sample sensitive attributes $G$. The objective function of our CFC consists of a self-supervised contrastive loss, a fair clustering loss, and a structural preservation loss.

*Self-supervised Contrastive Loss*. To learn a fair graph embedding using $X$, $S$, and $G$, we use a few components inspired by a recently proposed simple graph classification framework called Graph-MLP (Hu et al., 2021), which does not require message-passing between nodes and outperforms the classical message-passing GNN methods in various tasks (Wang et al., 2021; Yin et al., 2022). Specifically, it employs the neighboring contrastiveness and considers the $R$-hop neighbors to each node as the positive samples, and the remaining nodes as negative samples. The loss ensures that positive samples remain closer to the node, and negative samples remain farther away based on feature distance. Let $\gamma_{uv} = S_{uv}^R$ and $S$ is the co-association matrix, $sim$ denote cosine similarity, and $\tau$ be the temperature parameter, then we can write the loss as follows:

$$\mathcal{L}_c(Z, S) := -\frac{1}{n} \sum_{i=1}^n \log \frac{\sum_{a=1}^n \mathbb{1}_{[a \neq i]} \gamma_{ia} \exp(sim(Z_i, Z_a)/\tau)}{\sum_{b=1}^n \mathbb{1}_{[b \neq i]} \exp(sim(Z_i, Z_b)/\tau)}. \tag{2}$$

*Fair Clustering Loss*. Similar to other deep clustering approaches (Xie et al., 2016; Li et al., 2020), we employ a clustering assignment layer based on Student t-distribution and obtain soft cluster assignments $P$. We also include a fairness regularization term using an auxiliary target distribution $Q$

Table 2: Pre/post-attack performance when 15% group membership labels are switched.

| Algorithms | Metrics | *Office-31* | | | *MNIST-USPS* | | |
| --- | --- | --- | --- | --- | --- | --- | --- |
| | | Pre-Attack | Post Attack | Change (%) | Pre-Attack | Post Attack | Change (%) |
| CFC | Balance | $0.609 \pm 0.081$ | $0.606 \pm 0.047$ | (-)0.466 | $0.442 \pm 0.002$ | $0.373 \pm 0.090$ | (-)15.54 |
| | Entropy | $5.995 \pm 0.325$ | $6.009 \pm 0.270$ | (+)0.229 | $2.576 \pm 0.088$ | $2.626 \pm 0.136$ | (+)1.952 |
| | NMI | $0.690 \pm 0.019$ | $0.698 \pm 0.013$ | (+)1.236 | $0.269 \pm 0.007$ | $0.287 \pm 0.008$ | (+)6.749 |
| | ACC | $0.508 \pm 0.021$ | $0.511 \pm 0.018$ | (+)0.643 | $0.385 \pm 0.002$ | $0.405 \pm 0.015$ | (+)5.343 |
| SFD | Balance | $0.484 \pm 0.129$ | $0.062 \pm 0.080$ | (-)87.21 | $0.286 \pm 0.065$ | $0.000 \pm 0.000$ | (-)100.0 |
| | Entropy | $10.01 \pm 0.098$ | $9.675 \pm 0.187$ | (-)3.309 | $3.070 \pm 0.155$ | $1.621 \pm 0.108$ | (-)47.19 |
| | NMI | $0.801 \pm 0.050$ | $0.768 \pm 0.058$ | (-)4.110 | $0.320 \pm 0.033$ | $0.302 \pm 0.007$ | (-)5.488 |
| | ACC | $0.688 \pm 0.082$ | $0.624 \pm 0.098$ | (-)9.397 | $0.427 \pm 0.040$ | $0.378 \pm 0.015$ | (-)11.54 |
| FSC | Balance | $0.041 \pm 0.122$ | $0.000 \pm 0.000$ | (-)100.0 | $0.000 \pm 0.000$ | $0.000 \pm 0.000$ | (-)100.0 |
| | Entropy | $9.538 \pm 0.113$ | $9.443 \pm 0.178$ | (-)0.997 | $0.275 \pm 0.077$ | $0.251 \pm 0.041$ | (-)8.576 |
| | NMI | $0.669 \pm 0.014$ | $0.693 \pm 0.014$ | (+)3.659 | $0.549 \pm 0.011$ | $0.544 \pm 0.007$ | (-)0.807 |
| | ACC | $0.411 \pm 0.014$ | $0.452 \pm 0.027$ | (+)9.904 | $0.448 \pm 0.012$ | $0.457 \pm 0.002$ | (+)1.971 |
| KFC | Balance | $0.250 \pm 0.310$ | $0.057 \pm 0.172$ | (-)77.07 | $0.730 \pm 0.250$ | $0.352 \pm 0.307$ | (-)51.87 |
| | Entropy | $9.997 \pm 0.315$ | $9.919 \pm 0.189$ | (-)0.786 | $2.607 \pm 0.607$ | $2.343 \pm 0.444$ | (-)10.12 |
| | NMI | $0.393 \pm 0.064$ | $0.391 \pm 0.063$ | (-)0.483 | $0.072 \pm 0.024$ | $0.076 \pm 0.029$ | (+)5.812 |
| | ACC | $0.265 \pm 0.048$ | $0.266 \pm 0.049$ | (+)0.032 | $0.168 \pm 0.026$ | $0.176 \pm 0.034$ | (+)4.581 |

to ensure that the cluster assignments obtained from the learned embeddings $z \in Z$ are fair. We abuse notation slightly and denote the corresponding learned representation of sample $x \in X$ as $z_x \in Z$. Also let $p_k^x$ represent the probability of sample $x \in X$ being assigned to the $k$-th cluster, $\forall k \in [K]$. More precisely, $p_k^x$ represents the assigned confidence between representation $z_x$ and cluster centroid $c_k$ in the embedding space. The fair clustering loss term can then be written as:

$$\mathcal{L}_f(Z, G) := KL(P||Q) = \sum_{g \in [L]} \sum_{x \in H_g} \sum_{k \in [K]} p_k^x \log \frac{p_k^x}{q_k^x}, \tag{3}$$

where $p_k^x = \frac{(1+||z_x-c_k||^2)^{-1}}{\sum_{k' \in [K]}(1+||z_x-c_{k'}||^2)^{-1}}$ and $q_k^x = \frac{(p_k^x)^2/\sum_{x' \in H_{g(x)}} p_k^{x'}}{\sum_{k' \in [K]}(p_{k'}^x)^2/\sum_{x' \in H_{g(x)}} p_{k'}^{x'}}$.

*Structural Preservation Loss.* Since optimizing the fair clustering loss $\mathcal{L}_f$ can lead to a degenerate solution where the learned representation reduces to a constant function (Li et al., 2020), we employ a well-known structural preservation loss term for each protected group. Since this loss is applied to the final partitions obtained we omit it for clarity from Figure 3 which shows the internal CFC architecture. Let $P_g$ be the obtained soft cluster assignments for protected group $g$ using CFC, and $J_g$ be the cluster assignments for group $g$ obtained using any other well-performing fair clustering algorithm. We can then define this loss as originally proposed in Li et al. (2020):

$$\mathcal{L}_p := \sum_{g \in [L]} ||P_g P_g^\top - J_g J_g^\top||^2. \tag{4}$$

The overall objective for CFC algorithm can be written as $\mathcal{L}_c + \alpha \mathcal{L}_f + \beta \mathcal{L}_p$, where $\alpha, \beta$ are parameters used to control trade-off between individual losses. CFC can then be used to generate hard cluster label predictions using the soft cluster assignments $P \in \mathbb{R}^{n \times K}$.

## 3.2 RESULTS FOR THE DEFENSE

To showcase the efficacy of our CFC defense algorithm, we utilize the same datasets and fair clustering algorithms considered in the experiments for the attack section. Specifically, we show results when 15% of protected group membership labels can be switched for the adversary in Table 6 (over 10 individual runs). Here, we present pre-attack and post-attack fairness utility (Balance, Entropy) and clustering utility (NMI, ACC) values. We also denote the percent change in these evaluation metrics before the attack and after the attack for further analysis. The detailed implementation and hyperparameter choices for CFC can be found in Appendix F. The results for *Inverted UCI DIGITS* and *Extended Yale Face B* datasets are provided in Appendix G.

As can be seen in Table 6, our CFC clustering algorithm achieves fairness utility and clustering performance utility superlative to the other state-of-the-art fair clustering algorithms. In particular,

CFC does not optimize for only fairness utility over clustering performance utility but for both of these jointly, which is not the case for the other fair clustering algorithms. Post-attack performance of CFC on all datasets is also always better compared to the other fair algorithms where Balance often drops close to 0. This indicates that fairness utility has completely been disrupted for these algorithms and the adversary is successful. For CFC, post-attack fairness and performance values are at par with their pre-attack values, and at times even better than before the attack. For example, for Entropy, NMI, and ACC metrics, CFC has even better fairness and clustering performance after the attack than before it is undertaken on both the *Office-31* and *MNIST-USPS* datasets. Balance also decreases only by a marginal amount. Whereas for the other fair clustering algorithms SFD, FSC, and KFC, fairness has been completely disrupted through the poisoning attack. For all the other algorithms, Entropy and Balance decrease significantly with more than 10% and 85% decrease on average, respectively. Moreover, we provide a more in-depth analysis of CFC performance in Appendix H.

## 4 RELATED WORKS

**Fair Clustering**. Fair clustering aims to conduct unsupervised cluster analysis without encoding any bias into the instance assignments. Cluster fairness is evaluated using *Balance*, *i.e.*, to ensure that the size of sensitive demographic subgroups in any cluster follows the overall demographic ratio (Chierichetti et al., 2017). Some approaches use fairlets, where they decompose original data into multiple small and balanced partitions first, and use $k$-center or $k$-means clustering to get fair clusters (Schmidt et al., 2018; Backurs et al., 2019). There are also works that extend fair clustering into other clustering paradigms like spectral clustering (Kleindessner et al., 2019b), hierarchical clustering (Chhabra et al., 2020b), and deep clustering (Li et al., 2020; Wang & Davidson, 2019). In this work, beyond solutions to fair clustering, we investigate the vulnerabilities of fair clustering and corresponding defense algorithms.

**Adversarial Attacks on Clustering**. Recently, white-box and black-box adversarial attacks have been proposed against a number of different clustering algorithms (Chhabra et al., 2022a). For single-linkage hierarchical clustering, (Biggio et al., 2013) first proposed the *poisoning* and *obfuscation* attack settings, and provided algorithms that aimed to reduce clustering performance. In (Biggio et al., 2014) the authors extended this work to complete-linkage hierarchical clustering and (Crussell & Kegelmeyer, 2015) proposed a white-box poisoning attack for DBSCAN (Ester et al., 1996) clustering. On the other hand, (Chhabra et al., 2020a; Cinà et al., 2022) proposed black-box adversarial attacks that poison a small number of samples in the input data, so that when clustering is undertaken on the poisoned dataset, other unperturbed samples change cluster memberships, leading to a drop in overall clustering utility. As mentioned before, our attack is significantly different from these approaches, as it attacks the fairness utility of fair clustering algorithms instead of their clustering performance.

**Robustness of Fairness**. The robustness of fairness is the study of how algorithmic fairness could be violated or preserved under adversarial attacks or random perturbations. Solans et al. (2020) and Mehrabi et al. (2021b) propose poisoning attack frameworks trying to violate the predictive parity among subgroups in classification. Celis et al. (2021a) and Celis et al. (2021b) also work on classification and study fairness under perturbations on protected attributes. Differently, we turn our sights into an unsupervised scenario and study how attacks could degrade the fairness in clustering.

## 5 CONCLUSION

In this paper, we studied the fairness attack and defense problem. In particular, we proposed a novel black-box attack against fair clustering algorithms that works by perturbing a small percentage of samples' protected group memberships. For self-completeness, we also proposed a defense algorithm, named Consensus Fair Clustering (CFC), a novel fair clustering approach that utilizes consensus clustering along with fairness constraints to output robust and fair clusters. Conceptually, CFC combines consensus clustering with fair graph representation learning, which ensures that clusters are resilient to the fairness attack at both data and algorithm levels while possessing high clustering and fairness utility. Through extensive experiments on several real-world datasets using this fairness attack, we found that existing state-of-the-art fair clustering algorithms are highly susceptible to adversarial influence and their fairness utility can be reduced significantly. On the contrary, our proposed CFC algorithm is highly effective and robust as it resists the proposed fairness attack well.

## 6 ETHICS STATEMENT

In this paper, we have proposed a novel adversarial attack that aims to reduce the fairness utility of fair clustering models. Furthermore, we also propose a defense model based on consensus clustering and fair graph representation learning that is robust to the aforementioned attack. Both the proposed attack and the defense are important contributions that help facilitate ethics in ML research, due to a number of key reasons. First, there are very few studies that investigate the influence of adversaries on the fairness of unsupervised models (such as clustering), and hence, our work paves the way for future work that can study the effect of such attacks in other different learning models. Understanding the security vulnerabilities of models, especially with regard to fairness is the first step to developing models that are more robust to such attacks. Second, even though our attack approach can have a negative societal impact in the wrong hands, we propose a defense approach that can be used as a deterrent against such attacks. Third, we believe our defense approach can serve as a starting point for the development of more robust and fair ML models. Through this work, we seek to underscore the need for making fair clustering models robust to adversarial influence, and hence, drive the development of truly fair robust clustering models.

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

## A    Implementation of KFC, FSC, SFD Algorithms

We implemented the FSC, SFD, and KFC fair algorithms in Python, using the authors' implementations as a reference in case they were not written in Python. To this end, we generally default to using the hyperparameters for these algorithms as provided in the original implementations. However, if needed, we also tuned the hyperparameter values so as to maximize performance on the unsupervised fairness metrics (such as Balance) as this allows us to attack fairness better. Note that this is still an unsupervised parameter selection strategy as Balance is a fully unsupervised metric, as it takes only the clustering outputs and protected group memberships as input, which are also provided as input to the fair clustering algorithms.[2] Such parameter tuning has also been done in previous fair clustering work (Kleindessner et al., 2019b; Zhang & Davidson, 2021).

For SFD, we set the parameters $p = 2, q = 5$ for all datasets except *DIGITS* for which we set $p = 1, q = 5$. For FSC we use the default parameters and use the nearest neighbors approach Von Luxburg (2007) for creating the input graph for which we set the number of neighbors $= 3$ for all datasets. For KFC we use the default parameter value of $\delta = 0.1$.

## B    Definitions for Metrics

**NMI.** Normalized Mutual Information is essentially a normalized version of the widely used mutual information metric. Let $I$ denote the mutual information metric Shannon (1948), $E$ denote Shannon's entropy Shannon (1948), $L$ denote the cluster assignment labels, and $Y$ denote the ground truth labels. Then we can define NMI as:

$$\text{NMI} = \frac{I(Y, L)}{(1/2) * [E(Y) + E(L)]}.$$

**ACC.** This is the unsupervised equivalent of the traditional classification accuracy. Let there be a mapping function $\rho$ that computes all possible mappings between ground truth labels and possible cluster assignments[3] for some $m$ samples. Also, let $Y_i, L_i$ denote the ground truth label and cluster assignment label for the $i$-th sample, respectively. Then we can define ACC as:

$$\text{ACC} = \max_{\rho} \frac{\sum_{i=1}^{m} \mathbb{1}\{Y_i = \rho(L_i)\}}{m}.$$

**Balance.** Balance is a fairness metric proposed by (Chierichetti et al., 2017) which lies between 0 (least fair) and 1 (most fair). Let there be $m$ protected groups for a given dataset $X$. Then, define $r_X^g$ and $r_k^g$ to be the proportion of samples of the dataset belonging to protected group $g$ and the proportion of samples in cluster $k \in [K]$ belonging to protected group $g$. The Balance fairness notion is then defined over all clusters and protected groups as:

$$\text{Balance} = \min_{k \in [K], g \in [m]} \min\{\frac{r_X^g}{r_k^g}, \frac{r_k^g}{r_X^g}\}.$$

**Entropy.** Entropy is a fairness metric proposed by (Li et al., 2020) and similar to Balance, higher values of Entropy, mean that clusters have more fairness. Let $N_{k,g}$ be the set containing the samples of the dataset $X$ that belong to both the cluster $k \in [K]$ and the protected group $g$. Further, let $n_k$ be the number of samples in cluster $k$. Then Entropy for group $g$ is defined as follows:

$$\text{Entropy}(g) = -\sum_{k \in [K]} \frac{|N_{k,g}|}{n_k} \log \frac{|N_{k,g}|}{n_k}.$$

Note that in the paper, we take the average Entropy over all groups.

---

[2]Tuning hyperparameters using NMI/ACC or other performance metrics that take the ground truth cluster labels as input, would violate the unsupervised nature of the clustering problem.

[3]Such a mapping function can be computed optimally using the Hungarian assignment Kuhn (1955).

## C    STATISTICAL SIGNIFICANCE RESULTS

Table 3: KS test statistic values comparing our attack distribution with the random attack distribution for the Balance and Entropy metrics (** indicates statistical significance i.e., p-value < 0.01).

| Dataset | Algorithm | Balance | Entropy |
|---------|-----------|---------|---------|
| *Office-31* | SFD | 0.889** | 0.889** |
| *Office-31* | FSC | 0.889** | 0.778** |
| *Office-31* | KFC | 0.889** | 0.778** |
| *MNIST-USPS* | SFD | 0.222 | 0.222 |
| *MNIST-USPS* | FSC | 0.000 | 0.778** |
| *MNIST-USPS* | KFC | 0.333 | 0.333 |

We present the KS test statistic values in Table 3 for the SFD, FSC, and KFC fair clustering algorithms on the *Office-31* and *MNIST-USPS* datasets. The Balance and Entropy distributions obtained are largely significantly different except for the scenario when fairness utility for both our attack and the random attack quickly tends to 0. This leads to both distributions becoming identical, and no statistical test can be undertaken in that case. Moreover, it is important to note that such volatile performance is an artefact of the fair clustering algorithm, and does not relate to the attack approach.

## D    THEORETICAL RESULT FOR ATTACK

We present a simple result that demonstrates that an attacker solving our attack optimization can be successful at reducing fairness utility for a k-center or k-median fairlet decomposition based fair clustering model as described in (Chierichetti et al., 2017). We introduce notation used for this section independently below.

We will use an instance of well-separated ground-truth clusters as defined in (Chhabra et al., 2021b) with a slight modification allowing samples to possess protected group memberships, thus making it possible to study our attack. The original definition for well-separated clusters is as follows:

**Definition D.1** (Well-Separated Clusters (Chhabra et al., 2021b)). *These are defined for a given $K$ and dataset $X \in \mathbb{R}^{n \times d}$ as a set of cluster partitions $\{P_1, P_2, ..., P_K\}$ on the dataset, s.t. $|P_i| = n/K$ and points belonging to each $P_i \subset X$ are closer to each other than any points belonging to $P_j \subset X$ where $i \neq j, \forall i, j \in [K]$.*

We now provide a definition for well-separated ground-truth clusters with equitably distributed protected groups:

**Definition D.2** (Well-Separated Clusters with Equitable Group Distribution). *These are defined for a given $K$, dataset $X \in \mathbb{R}^{n \times d}$, and protected group memberships $G \in [L]^n \in \mathbb{N}^n$, as a set of cluster partitions $\{P_1, P_2, ..., P_K\}$ on the dataset, s.t. $|P_i| = n/K$ and points belonging to each $P_i \subset X$ are closer to each other than any points belonging to $P_j \subset X$ and $P_i$ contains an equal number of protected group members of $G$ as $P_j$ without overlap, where $i \neq j, \forall i, j \in [K]$.*

Using the Definition D.2 provided above for well-separated clusters with equitable protected groups, we will now provide a result that proves the success of our attack for the specific case when $K = 2$ and $L = 2$ (i.e. we have two protected groups).

**Theorem D.1.** *Given a ground-truth instance as defined in Definition D.2 for $K = 2, L = 2$, Fairlet Decomposition (Chierichetti et al., 2017) as the clustering model, and an attacker that satisfies the following conditions: (i) controls equal number of samples from each protected group in each ground-truth cluster, and (ii) has selected these points s.t. they are clustered similarly before and after the attack; then our attack optimization will always be successful at reducing the Balance of the benign samples after the attack.*

*Proof.* Based on Definition D.2 we know that we have two ground truth clusters $P_1$ and $P_2$ each containing $n/2$ samples. Following Definition D.2, let the two protected groups be denoted as $g_1$ and $g_2$ and then we know there are $n/4$ samples of $g_1$ and $n/4$ samples of $g_2$ in both $P_1$ and $P_2$,

individually. The attacker controls a total of $A$ points and if the first condition of the theorem is met, they control $A/2$ points in each ground-truth cluster comprising of $A/4$ points from each of the two protected groups.

First note that the Balance of the overall dataset including the adversary's points is clearly 1.0. The defender will use this value for the fair clustering model since they do not know the attacker's points from benign points. Hence, the Fairlet Decomposition algorithm is invoked with $= 1.0$, specifying a requirement of having Balance in each cluster as 1.0. For this case the Bipartite Graph Matching approach is used for fairlet decomposition (refer to (Chierichetti et al., 2017) for more details).

Note that when Fairlet Decomposition is run before the attacker switches memberships and carries out the attack, the ground truth clusters are also the optimal fair clustering solution. We can then calculate the Balance for just the benign points, denoted as $\phi_{pre}$. Due to symmetry it will be the same for each protected group in each cluster:

$$\phi_{pre} = \frac{n/4 - A/4}{n/2} \times \frac{n}{n/2 - A/2} = 1.$$

Thus, $\phi_{pre}$ is the maximum possible value of Balance possible, and if we can show that an attack solution exists that switches group memberships such that post-attack Balance for the benign samples $\phi_{post} < \phi_{pre} = 1$ the theorem statement will hold true.

Consider the following feasible attack policy: the attacker switches membership of each of the $A/4$ points of group $g_1$ in $P_1$ to group $g_2$ and switches membership of each of the $A/4$ points of group $g_2$ in $P_2$ to group $g_1$. Now, note that after carrying out the attack Balance of the overall dataset is still 1.0 and hence, the same algorithm is going to be invoked for fair clustering. After the attack, however, $P_1$ and $P_2$ are no longer the optimal clusters as they violate the Balance constraint. Hence, the optimal clusters will be $P_1'$ and $P_2'$ where $|P_1'| = |P_2'| = n/2$. $P_1'$ has obtained $A/4$ group $g_1$ points from $P_2$ to accommodate the Balance constraint. Similarly, $P_2'$ has obtained $A/4$ group $g_2$ points from $P_1$.

Now, if the second condition of the theorem holds, without loss of generality, the original $A/2$ adversarial points of $P_1$ are still present in $P_1'$, and similarly the original $A/2$ adversarial points of $P_2$ are present in $P_2'$. Again, note that the attack policy was such that the $A/2$ points of $P_1$ now all belong to protected group $g_2$ and the $A/2$ points of $P_2$ now all belong to protected group $g_1$. Therefore, we can calculate the Balance for the benign samples post the attack:

$$\begin{aligned}
\phi_{post} &= \frac{n/4 - A/2}{n/2} \times \frac{n}{n/2 - A/2} \\
&= \frac{n - 2A}{n - A} \\
&= \frac{n - A}{n - A} - \frac{A}{n - A} \\
&= 1 - \frac{A}{n - A} \\
&< 1 = \phi_{pre}
\end{aligned}$$

Thus, we have proved that $\phi_{post} < \phi_{pre}$ for all $A > 0$.

$\square$

# E    ATTACK RESULTS ON DIGITS AND YALE DATASETS

We present results obtained upon carrying out our attack and the random attack on the SFD, FSC, and KFC fair algorithms and two additional datasets: *Extended Yale Face B* (Lee et al., 2005) (abbreviated as *Yale*) and *Inverted UCI DIGITS* (Xu et al., 1992) (abbreviated as *DIGITS*). For *Yale*, we consider lighting and elevation as the two sensitive attributes, and for *DIGITS*, we essentially color invert all the images in the original *DIGITS* dataset and then consider the sensitive attribute to be the source of the image (inverted or original). The attack analysis that follows in Figure 4 is exactly similar to the results on *Office-31* and *MNIST-USPS* in the main paper. We can see that our attack is significantly more detrimental than the random attack, and reduces fairness utility by a large margin compared to the pre-attack values.

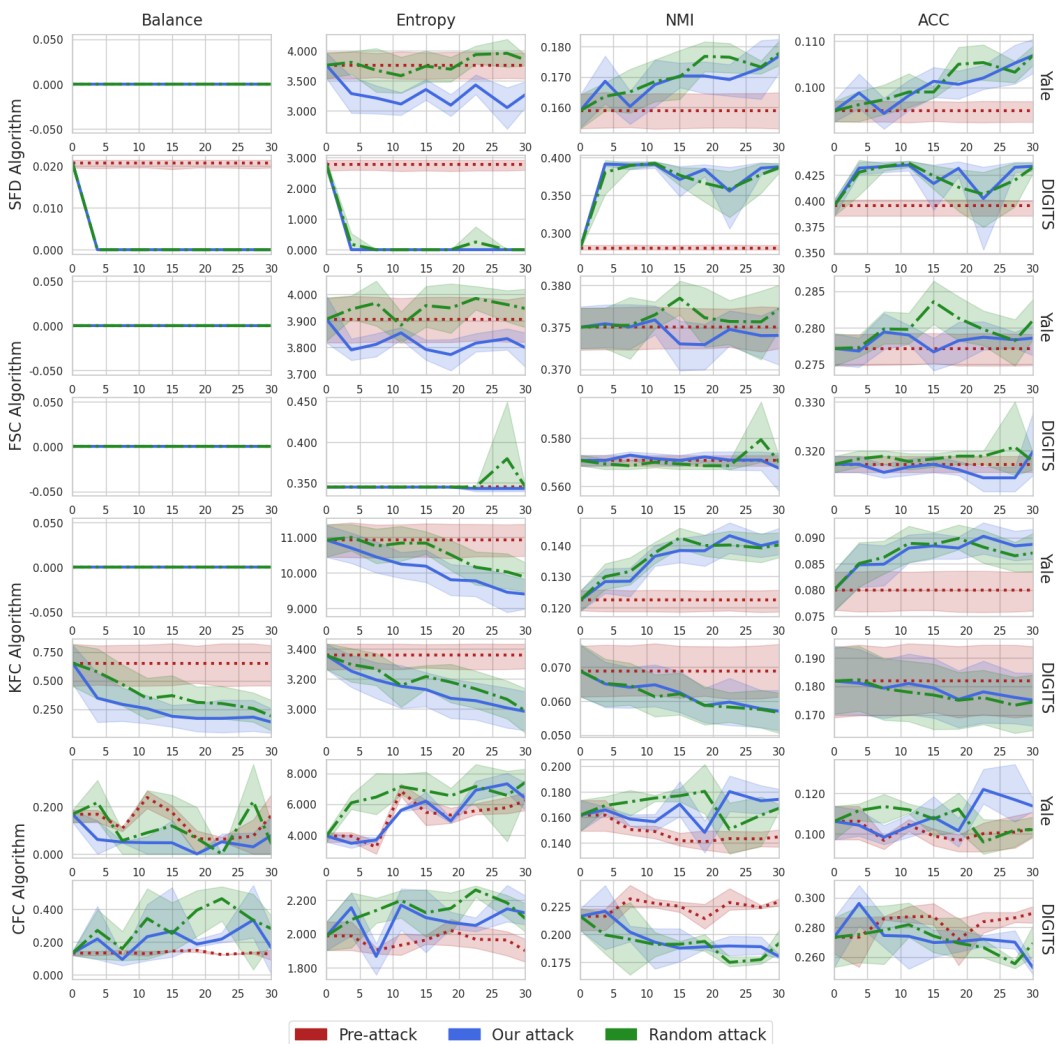

Figure 4: Attack results for the *Extended Yale Face B* and *Inverted UCI DIGITS* datasets (x-axis denotes % of samples attacker can poison).

# F  IMPLEMENTATION OF CFC

Here we present parameter values and implementation details regarding CFC. The hyperparameters such as number of basic partitions $r$, temperature parameter $\tau$ in the contrastive loss $\mathcal{L}_c$, dropout in hidden layers, number of training epochs, the activation function, and fair clustering algorithm used to generate $J$ for structural preservation loss $\mathcal{L}_p$ are set to be the same across all datasets. These are $r = 100, \tau = 2, \text{dropout} = 0.6, \# \text{epochs} = 3000$, Gaussian Error Linear Unit (Hendrycks & Gimpel, 2016) is used as the activation function, and we use SFD with default parameters for generating $J$ since it runs faster than other fair clustering algorithms. Moreover, the dimension of the hidden layer is set to 256 for all datasets except for *DIGITS* since *DIGITS* has only 64 features and hence we use the hidden layer dimension as 36 for it.

As mentioned in Appendix A, we tune the other hyperparameters for the different datasets to optimize for fairness performance. Using grid based search we set the following parameters for the given datasets: for *Office-31* we have $R = 1, \alpha = 1, \beta = 100$; for *MNIST-USPS* we have $R = 2, \alpha = 100, \beta = 25$; for *Yale* we have $R = 2, \alpha = 50, \beta = 10$; and for *DIGITS* we have $R = 2, \alpha = 10, \beta = 50$.

Table 4: Pre/post-attack performance when 15% group membership labels are switched.

| Algorithms | Metrics | DIGITS | | | Yale | | |
|---|---|---|---|---|---|---|---|
| | | Pre-Attack | Post Attack | Change (%) | Pre-Attack | Post Attack | Change (%) |
| CFC | Balance | $0.145 \pm 0.002$ | $0.266 \pm 0.189$ | (+)83.62 | $0.176 \pm 0.027$ | $0.047 \pm 0.067$ | (-)73.22 |
| | Entropy | $1.963 \pm 0.022$ | $2.096 \pm 0.153$ | (+)6.758 | $5.472 \pm 0.669$ | $6.196 \pm 0.441$ | (+)13.22 |
| | NMI | $0.225 \pm 0.004$ | $0.187 \pm 0.013$ | (-)16.72 | $0.142 \pm 0.004$ | $0.170 \pm 0.013$ | (+)20.16 |
| | ACC | $0.287 \pm 0.006$ | $0.270 \pm 0.008$ | (-)6.132 | $0.099 \pm 0.003$ | $0.108 \pm 0.007$ | (+)9.381 |
| SFD | Balance | $0.021 \pm 0.002$ | $0.000 \pm 0.000$ | (-)100.0 | $0.000 \pm 0.000$ | $0.000 \pm 0.000$ | (-)100.0 |
| | Entropy | $2.781 \pm 0.286$ | $0.000 \pm 0.000$ | (-)100.0 | $3.757 \pm 0.358$ | $3.351 \pm 0.277$ | (-)10.81 |
| | NMI | $0.281 \pm 0.005$ | $0.371 \pm 0.032$ | (+)32.21 | $0.159 \pm 0.009$ | $0.170 \pm 0.007$ | (+)7.206 |
| | ACC | $0.395 \pm 0.014$ | $0.417 \pm 0.038$ | (+)5.440 | $0.095 \pm 0.004$ | $0.101 \pm 0.005$ | (+)6.733 |
| FSC | Balance | $0.000 \pm 0.000$ | $0.000 \pm 0.000$ | (-)100.0 | $0.000 \pm 0.000$ | $0.000 \pm 0.000$ | (-)100.0 |
| | Entropy | $0.345 \pm 0.000$ | $0.345 \pm 0.000$ | (-)0.000 | $3.907 \pm 0.137$ | $3.793 \pm 0.060$ | (-)2.912 |
| | NMI | $0.571 \pm 0.004$ | $0.571 \pm 0.004$ | (-)0.000 | $0.375 \pm 0.004$ | $0.373 \pm 0.005$ | (-)0.530 |
| | ACC | $0.317 \pm 0.003$ | $0.317 \pm 0.003$ | (-)0.000 | $0.277 \pm 0.004$ | $0.277 \pm 0.004$ | (-)0.171 |
| KFC | Balance | $0.653 \pm 0.290$ | $0.188 \pm 0.167$ | (-)71.14 | $0.000 \pm 0.000$ | $0.000 \pm 0.000$ | (-)100.0 |
| | Entropy | $3.356 \pm 0.138$ | $3.128 \pm 0.193$ | (-)6.815 | $10.92 \pm 0.744$ | $10.19 \pm 0.742$ | (-)6.730 |
| | NMI | $0.069 \pm 0.012$ | $0.062 \pm 0.010$ | (-)9.184 | $0.122 \pm 0.006$ | $0.138 \pm 0.006$ | (+)13.08 |
| | ACC | $0.182 \pm 0.020$ | $0.179 \pm 0.017$ | (-)1.333 | $0.080 \pm 0.006$ | $0.088 \pm 0.004$ | (+)10.50 |

Figure 5: (A) Histogram representing the distribution of Balance of the Basic Partitions (BPs) before the attack and after the attack. (B) Visualization of the consensus matrix before the attack. (C) Visualization of the conensus matrix after the attack.

## G  DEFENSE RESULTS ON DIGITS AND YALE DATASETS

Table 4 shows the pre/post-attack performance of CFC and other fair clustering algorithms on *DIGITS* and *Yale* when 15% group membership labels are switched. As can be seen, CFC is superlative in terms of both pre-attack and post-attack fairness utility and clustering performance compared to the SFD, FSC, and KFC algorithms. More importantly, note that for the *Yale* dataset, the Balance is consistently 0.0 for all three competitive algorithms, leading to a 100% decrease in fairness (both pre-attack and post-attack Balance is 0). However, CFC is able to find a clustering solution that has non-zero pre-attack Balance and even though there is a drop in Balance after the attack, it never reaches 0. In fact, Entropy increases by 13.22%. For *DIGITS*, its results are even better, and CFC has a good trade-off between both clustering utility and fairness utility. Specifically, after the attack, Balance for CFC increases by 83.62% and Entropy by 6.758%. For all other state-of-the-art algorithms, performance decreases significantly for both Balance and Entropy.

## H  IN-DEPTH EXPLORATION OF CFC

### H.1  ANALYZING THE CONSENSUS CLUSTERING STAGE OF CFC

We undertake some additional analysis that sheds light on why the performance of CFC remains largely unaffected by the proposed fairness attack. We begin by analyzing the first stage of the CFC pipeline, i.e., the consensus matrix generation stage. In Figure 5(A), we plot the distribution of basic partitions' (BPs) Balance values before our fairness attack and after our fairness attack on *Office-31*, as a histogram. Note that $r = 100$, which means that we have 100 basic partitions. It can be seen

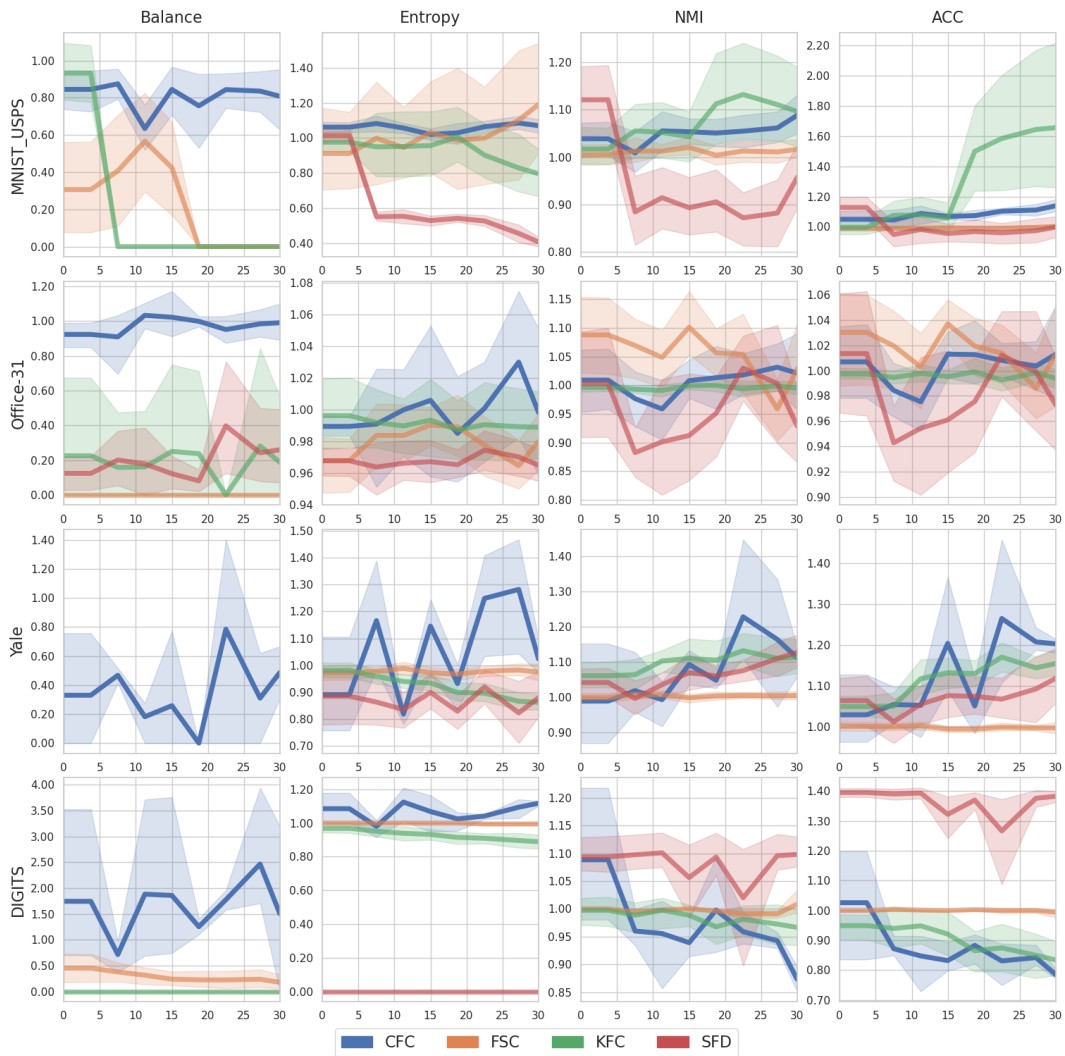

Figure 6: Post/Pre attack ratio trends for CFC and other fair clustering algorithms (we do not plot curves for which pre-attack values are 0). X-axis denotes the % of samples attacker can poison.

that before the attack there are more BPs with 0 Balance values, and after the attack these partitions actually decrease. Specifically, the mean Balance of the BPs shifts from 0.3 before the attack to 0.35 after the attack. Moreover, the partition at the 20th percentile has Balance 0 before the attack, but 20th percentile BP improves to a Balance of 0.14 after the attack. This indicates that the simple basic partition generation strategy will alleviate the negative impact of a fairness attack.

Moreover, it is beneficial to use consensus between BPs as a means of ensuring robustness, i.e, our model is able to generalize from the performance of a number of different clustering results to obtain more robust results. This can also be observed through Figures 5(B) and 5(C), where we plot the pre-attack and post-attack consensus matrices obtained for *Office-31*, respectively. Visually, both these matrices look similar, indicating that the consensus matrix is largely unaffected by the attack. Note that for the size of these matrices ($n \times n$, where $n = 1,293$ is the number of samples in the *Office-31* dataset), the norm of their difference equals to 19.335, which is relatively small comparing the number of samples. This indicates that consensus clustering results as part of the first stage are independent of the attack, and hence, can be used to ensure highly resilient and robust performance on the dataset before and after the attack.

## H.2 ANALYZING OVERALL ADVERSARIAL ROBUSTNESS OF CFC

Next, we conduct experiments on comparing the performance of CFC with the other state-of-the-art fair clustering algorithms. For ease of understanding, we plot the ratio between the mean post-attack and pre-attack values as a function of the percentage of protected group membership labels switched by the attacker. Thus, the ratio is mathematically defined as $\frac{\text{Mean Post-Attack Value}}{\text{Mean Pre-Attack Value}}$. Then, note that higher values of the ratio, indicate more robust performance with regards to fairness metrics such as Balance or Entropy.

We present these results in Figure 6. As can be observed, the CFC ratio values are always much higher than the other algorithms for all the attack percentages and for the fairness metrics Balance and Entropy. This is especially observable for certain datasets (such as Yale), where Balance for all other fair algorithms is consistently 0, but CFC is still able to obtain clustering solutions with desirable fairness utility before and after the attack. It is also worthwhile to note that for *Office-31* and *MNIST-USPS*, fairness performance is highly robust as the ratio trends tend to be approximately 1.0, or >1.0, with little to no decrease in utility after the attack. For the NMI and ACC metrics, we find that the ratio is generally distributed very close to 1.0, indicating that clustering performance is very similar before and after the attack. This means that in general, it is hard to tell whether or not a fairness attack has occurred based on clustering performance. This makes it challenging for the defender to defend against such an attack, further mandating the need for robust fair clustering algorithms like CFC. Also note that for some algorithms, pre-attack and/or post-attack values are consistently 0, and we omit their trends from the figure since they are indeterminate.

# I MISCELLANEOUS EXPERIMENTS AND RESULTS

## I.1 RESULTS WHEN ATTACKER CAN SWITCH UP TO 3.75% GROUP MEMBERSHIPS

Table 5: Results for our attack and random attack when 3.75% group membership labels are switched.

| Algorithms | Metrics | Office-31 | | | | |
|---|---|---|---|---|---|---|
| | | Pre-Attack | Our Attack | Change (%) | Random Attack | Change (%) |
| SFD | Balance | $0.484 \pm 0.129$ | $0.068 \pm 0.089$ | (-)85.95 | $0.250 \pm 0.138$ | (-)48.37 |
| | Entropy | $10.01 \pm 0.098$ | $9.681 \pm 0.161$ | (-)3.243 | $9.798 \pm 0.146$ | (-)2.082 |
| | NMI | $0.801 \pm 0.050$ | $0.809 \pm 0.040$ | (+)0.956 | $0.802 \pm 0.060$ | (+)0.173 |
| | ACC | $0.688 \pm 0.082$ | $0.681 \pm 0.066$ | (-)1.086 | $0.673 \pm 0.101$ | (-)2.196 |
| FSC | Balance | $0.041 \pm 0.122$ | $0.000 \pm 0.000$ | (-)100.0 | $0.032 \pm 0.097$ | (-)20.95 |
| | Entropy | $9.538 \pm 0.113$ | $9.228 \pm 0.191$ | (-)3.257 | $9.322 \pm 0.176$ | (-)2.268 |
| | NMI | $0.669 \pm 0.014$ | $0.688 \pm 0.028$ | (+)2.920 | $0.687 \pm 0.021$ | (+)2.714 |
| | ACC | $0.411 \pm 0.014$ | $0.446 \pm 0.034$ | (+)8.525 | $0.446 \pm 0.030$ | (+)8.420 |
| KFC | Balance | $0.250 \pm 0.310$ | $0.052 \pm 0.155$ | (-)79.36 | $0.170 \pm 0.267$ | (-)32.08 |
| | Entropy | $9.997 \pm 0.315$ | $9.947 \pm 0.146$ | (-)0.500 | $9.966 \pm 0.251$ | (-)0.313 |
| | NMI | $0.393 \pm 0.064$ | $0.392 \pm 0.063$ | (-)0.272 | $0.393 \pm 0.064$ | (-)0.149 |
| | ACC | $0.265 \pm 0.048$ | $0.265 \pm 0.050$ | (-)0.227 | $0.266 \pm 0.048$ | (+)0.065 |
| Algorithms | Metrics | MNIST-USPS | | | | |
| | | Pre-Attack | Our Attack | Change (%) | Random Attack | Change (%) |
| SFD | Balance | $0.286 \pm 0.065$ | $0.252 \pm 0.015$ | (-)12.05 | $0.425 \pm 0.146$ | (+)48.45 |
| | Entropy | $3.070 \pm 0.155$ | $3.101 \pm 0.073$ | (+)1.012 | $3.206 \pm 0.090$ | (+)4.432 |
| | NMI | $0.320 \pm 0.033$ | $0.358 \pm 0.025$ | (+)11.73 | $0.328 \pm 0.030$ | (+)2.503 |
| | ACC | $0.427 \pm 0.040$ | $0.475 \pm 0.025$ | (+)11.15 | $0.444 \pm 0.033$ | (+)4.112 |
| FSC | Balance | $0.000 \pm 0.000$ | $0.000 \pm 0.000$ | (-)100.0 | $0.000 \pm 0.000$ | (-)100.0 |
| | Entropy | $0.275 \pm 0.077$ | $0.230 \pm 0.067$ | (-)16.17 | $0.276 \pm 0.063$ | (+)0.564 |
| | NMI | $0.549 \pm 0.011$ | $0.542 \pm 0.010$ | (-)1.314 | $0.549 \pm 0.009$ | (-)0.022 |
| | ACC | $0.448 \pm 0.012$ | $0.450 \pm 0.013$ | (+)0.461 | $0.452 \pm 0.009$ | (+)0.847 |
| KFC | Balance | $0.730 \pm 0.250$ | $0.225 \pm 0.298$ | (-)69.14 | $0.414 \pm 0.359$ | (-)43.37 |
| | Entropy | $2.607 \pm 0.607$ | $2.502 \pm 0.466$ | (-)4.021 | $2.710 \pm 0.464$ | (+)3.978 |
| | NMI | $0.072 \pm 0.024$ | $0.072 \pm 0.024$ | (+)0.110 | $0.073 \pm 0.023$ | (+)0.948 |
| | ACC | $0.168 \pm 0.026$ | $0.171 \pm 0.025$ | (+)1.453 | $0.169 \pm 0.028$ | (+)0.715 |

## I.2 UPDATED TABLES WITH RANDOM ATTACK

Table 6: Pre/post-attack performance when 15% group membership labels are switched for *Office-31* and *MNIST-USPS*.

| Algorithms | Metrics | Office-31 | | | | | MNIST-USPS | | | | |
|---|---|---|---|---|---|---|---|---|---|---|---|
| | | Pre-Attack | Our Attack | Change (%) | Random Attack | Change (%) | Pre-Attack | Our Attack | Change (%) | Random Attack | Change (%) |
| CFC | Balance | 0.609 ± 0.081 | 0.606 ± 0.047 | (-)0.466 | 0.580 ± 0.083 | (-)4.741 | 0.442 ± 0.002 | 0.373 ± 0.09 | (-)15.54 | 0.298 ± 0.125 | (-)32.59 |
| | Entropy | 5.995 ± 0.325 | 6.009 ± 0.270 | (+)0.229 | 5.969 ± 0.116 | (-)0.436 | 2.576 ± 0.088 | 2.626 ± 0.136 | (+)1.952 | 2.559 ± 0.138 | (-)0.680 |
| | NMI | 0.690 ± 0.019 | 0.698 ± 0.013 | (+)1.236 | 0.715 ± 0.007 | (+)3.621 | 0.269 ± 0.007 | 0.287 ± 0.008 | (+)6.749 | 0.280 ± 0.003 | (+)4.007 |
| | ACC | 0.508 ± 0.021 | 0.511 ± 0.018 | (+)0.643 | 0.527 ± 0.013 | (+)3.765 | 0.385 ± 0.002 | 0.405 ± 0.015 | (+)5.343 | 0.394 ± 0.005 | (+)2.439 |
| SFD | Balance | 0.484 ± 0.129 | 0.062 ± 0.080 | (-)87.21 | 0.212 ± 0.188 | (-)56.34 | 0.286 ± 0.065 | 0.000 ± 0.000 | (-)100.0 | 0.000 ± 0.000 | (-)100.0 |
| | Entropy | 10.01 ± 0.098 | 9.675 ± 0.187 | (-)3.309 | 9.748 ± 0.181 | (-)2.581 | 3.070 ± 0.155 | 1.621 ± 0.108 | (-)47.19 | 1.743 ± 0.156 | (-)43.23 |
| | NMI | 0.801 ± 0.050 | 0.768 ± 0.058 | (-)4.110 | 0.795 ± 0.053 | (-)0.726 | 0.320 ± 0.033 | 0.302 ± 0.007 | (-)5.488 | 0.301 ± 0.019 | (-)5.847 |
| | ACC | 0.688 ± 0.082 | 0.624 ± 0.098 | (-)9.397 | 0.668 ± 0.091 | (-)2.908 | 0.427 ± 0.040 | 0.378 ± 0.015 | (-)11.54 | 0.374 ± 0.026 | (-)12.35 |
| FSC | Balance | 0.041 ± 0.122 | 0.000 ± 0.000 | (-)100.0 | 0.086 ± 0.172 | (+)110.8 | 0.000 ± 0.000 | 0.000 ± 0.000 | (-)100.0 | 0.000 ± 0.000 | (-)0.000 |
| | Entropy | 9.538 ± 0.113 | 9.443 ± 0.178 | (-)0.997 | 9.558 ± 0.226 | (+)0.207 | 0.275 ± 0.077 | 0.251 ± 0.041 | (-)8.576 | 0.295 ± 0.036 | (+)7.598 |
| | NMI | 0.669 ± 0.014 | 0.693 ± 0.014 | (+)3.659 | 0.693 ± 0.022 | (+)3.711 | 0.549 ± 0.011 | 0.544 ± 0.007 | (-)0.807 | 0.552 ± 0.006 | (+)0.491 |
| | ACC | 0.411 ± 0.014 | 0.452 ± 0.027 | (+)9.904 | 0.447 ± 0.030 | (+)8.817 | 0.448 ± 0.012 | 0.457 ± 0.002 | (+)1.971 | 0.455 ± 0.002 | (+)1.510 |
| KFC | Balance | 0.250 ± 0.310 | 0.057 ± 0.172 | (-)77.07 | 0.194 ± 0.301 | (-)22.55 | 0.730 ± 0.250 | 0.352 ± 0.307 | (-)51.87 | 0.608 ± 0.237 | (-)16.72 |
| | Entropy | 9.997 ± 0.315 | 9.919 ± 0.189 | (-)0.786 | 9.992 ± 0.250 | (-)0.051 | 2.607 ± 0.607 | 2.343 ± 0.444 | (-)10.12 | 2.383 ± 0.490 | (-)8.595 |
| | NMI | 0.393 ± 0.064 | 0.391 ± 0.063 | (-)0.483 | 0.393 ± 0.067 | (-)0.160 | 0.072 ± 0.024 | 0.076 ± 0.029 | (+)5.812 | 0.076 ± 0.027 | (+)5.330 |
| | ACC | 0.265 ± 0.048 | 0.266 ± 0.049 | (+)0.032 | 0.265 ± 0.051 | (-)0.162 | 0.168 ± 0.026 | 0.176 ± 0.034 | (+)4.581 | 0.174 ± 0.031 | (+)3.419 |

Table 7: Pre/post-attack performance when 15% group membership labels are switched for *DIGITS* and *Yale*.

| Algorithms | Metrics | DIGITS | | | | | Yale | | | | |
|---|---|---|---|---|---|---|---|---|---|---|---|
| | | Pre-Attack | Our Attack | Change (%) | Random Attack | Change (%) | Pre-Attack | Our Attack | Change (%) | Random Attack | Change (%) |
| CFC | Balance | 0.145 ± 0.002 | 0.266 ± 0.189 | (+)83.62 | 0.252 ± 0.134 | (+)74.33 | 0.176 ± 0.027 | 0.047 ± 0.067 | (-)73.22 | 0.119 ± 0.102 | (-)32.49 |
| | Entropy | 1.963 ± 0.022 | 2.096 ± 0.153 | (+)6.758 | 2.125 ± 0.114 | (+)8.244 | 5.472 ± 0.669 | 6.196 ± 0.441 | (+)13.22 | 6.864 ± 0.892 | (+)25.42 |
| | NMI | 0.225 ± 0.004 | 0.187 ± 0.013 | (-)16.72 | 0.191 ± 0.003 | (-)15.14 | 0.142 ± 0.004 | 0.170 ± 0.013 | (+)20.16 | 0.177 ± 0.004 | (+)24.84 |
| | ACC | 0.287 ± 0.006 | 0.270 ± 0.008 | (-)6.132 | 0.274 ± 0.009 | (-)4.610 | 0.099 ± 0.003 | 0.108 ± 0.007 | (+)9.381 | 0.107 ± 0.006 | (+)8.583 |
| SFD | Balance | 0.021 ± 0.002 | 0.000 ± 0.000 | (-)100.0 | 0.000 ± 0.000 | (-)100.0 | 0.000 ± 0.000 | 0.000 ± 0.000 | (-)100.0 | 0.000 ± 0.000 | (-)100.0 |
| | Entropy | 2.781 ± 0.286 | 0.000 ± 0.000 | (-)100.0 | 0.000 ± 0.000 | (-)100.0 | 3.757 ± 0.358 | 3.351 ± 0.277 | (-)10.81 | 3.741 ± 0.432 | (-)0.430 |
| | NMI | 0.281 ± 0.005 | 0.371 ± 0.032 | (+)32.21 | 0.377 ± 0.020 | (+)34.18 | 0.159 ± 0.009 | 0.170 ± 0.007 | (+)7.206 | 0.170 ± 0.013 | (+)7.053 |
| | ACC | 0.395 ± 0.014 | 0.417 ± 0.038 | (+)5.440 | 0.424 ± 0.019 | (+)7.280 | 0.095 ± 0.004 | 0.101 ± 0.005 | (+)6.733 | 0.099 ± 0.002 | (+)4.302 |
| FSC | Balance | 0.000 ± 0.000 | 0.000 ± 0.000 | (-)100.0 | 0.000 ± 0.000 | (-)100.0 | 0.000 ± 0.000 | 0.000 ± 0.000 | (-)100.0 | 0.000 ± 0.000 | (-)100.0 |
| | Entropy | 0.345 ± 0.000 | 0.345 ± 0.000 | (-)0.000 | 0.345 ± 0.000 | (+)0.020 | 3.907 ± 0.137 | 3.793 ± 0.060 | (-)2.912 | 3.958 ± 0.125 | (+)1.304 |
| | NMI | 0.571 ± 0.004 | 0.571 ± 0.004 | (-)0.000 | 0.569 ± 0.003 | (-)0.258 | 0.375 ± 0.004 | 0.373 ± 0.005 | (-)0.530 | 0.378 ± 0.003 | (+)0.922 |
| | ACC | 0.317 ± 0.003 | 0.317 ± 0.003 | (-)0.000 | 0.318 ± 0.003 | (+)0.351 | 0.277 ± 0.004 | 0.277 ± 0.004 | (-)0.171 | 0.284 ± 0.005 | (+)2.306 |
| KFC | Balance | 0.653 ± 0.290 | 0.188 ± 0.167 | (-)71.14 | 0.369 ± 0.290 | (-)43.47 | 0.000 ± 0.000 | 0.000 ± 0.000 | (-)100.0 | 0.000 ± 0.000 | (-)100.0 |
| | Entropy | 3.356 ± 0.138 | 3.128 ± 0.193 | (-)6.815 | 3.215 ± 0.135 | (-)4.216 | 10.92 ± 0.744 | 10.19 ± 0.742 | (-)6.730 | 10.84 ± 0.512 | (-)0.785 |
| | NMI | 0.069 ± 0.012 | 0.062 ± 0.010 | (-)9.184 | 0.062 ± 0.010 | (-)9.749 | 0.122 ± 0.006 | 0.138 ± 0.006 | (+)13.08 | 0.142 ± 0.006 | (+)16.42 |
| | ACC | 0.182 ± 0.020 | 0.179 ± 0.017 | (-)1.333 | 0.177 ± 0.018 | (-)2.710 | 0.080 ± 0.006 | 0.088 ± 0.004 | (+)10.50 | 0.089 ± 0.004 | (+)10.87 |

## I.3 CFC PERFORMANCE WHEN $J$ IS GENERATED USING DIFFERENT FAIR ALGORITHMS

Table 8: CFC performance on *Office-31* for $J$ generated by different algorithms.

| Algorithm used for $J$ | Metrics | Pre-Attack | Post-Attack | Change (%) |
|---|---|---|---|---|
| SFD | Balance | 0.609 ± 0.081 | 0.606 ± 0.047 | (-)0.466 |
| | Entropy | 5.995 ± 0.325 | 6.009 ± 0.270 | (+)0.229 |
| | NMI | 0.690 ± 0.019 | 0.698 ± 0.013 | (+)1.236 |
| | ACC | 0.508 ± 0.021 | 0.511 ± 0.018 | (+)0.643 |
| FSC | Balance | 0.650 ± 0.069 | 0.653 ± 0.064 | (+)0.404 |
| | Entropy | 4.027 ± 0.273 | 4.485 ± 0.677 | (+)11.38 |
| | NMI | 0.610 ± 0.008 | 0.610 ± 0.017 | (-)0.011 |
| | ACC | 0.361 ± 0.015 | 0.358 ± 0.015 | (-)0.714 |
| KFC | Balance | 0.414 ± 0.318 | 0.385 ± 0.310 | (-)6.815 |
| | Entropy | 4.830 ± 0.174 | 4.382 ± 0.298 | (-)9.260 |
| | NMI | 0.409 ± 0.003 | 0.426 ± 0.004 | (+)4.218 |
| | ACC | 0.252 ± 0.001 | 0.265 ± 0.002 | (+)5.108 |

