# OpenReview forum: "Robust Fair Clustering: A Novel Fairness Attack and Defense Framework"
_ICLR.cc/2023/Conference — ICLR 2023 poster_

### Official Review · Reviewer_EUnH · 2022-10-24

**Confidence:** 1
**Correctness:** 3
**Technical Novelty And Significance:** 3
**Empirical Novelty And Significance:** 3
**Recommendation:** 6

**Clarity, Quality, Novelty And Reproducibility:**

The issue of robustness under attack is well-motivated. The designed attack is quite novel.

**Strength And Weaknesses:**

Strength:
1. The issue of robustness under attack is well-motivated. I believe this could be an important property for any robust fair clustering.
2. The designed attack is quite novel.

My major concern is the following:
The proposed Consensus Fair Clustering appears to respond well to the designed attack. Is there any guarantee that it remains robust under ANY attack that the adversary would choose?

**Summary Of The Paper:**

The paper studies robust fair clustering and motivates the issue of robustness under attack. The authors conduct numerical experiments to show that state-of-the-art models are highly susceptible to the attack they design. The authors then propose a Consensus Fair Clustering and numerically show its robustness.

**Summary Of The Review:**

The issue of robustness under attack is well-motivated, and I believe this could be an important property for any robust fair clustering.
2. The designed attack is quite novel. The designed attack is quite novel.

---

> ### Author Response · Authors · 2022-11-16
> **Response to Reviewer EUnH**
>
> We thank Reviewer EUnH for their review, the insightful feedback received, and appreciate the time spent in reviewing our work. We have provided responses to the concerns below.
>
> **ANY attack that the adversary would choose**. Since CFC (and any other fair clustering algorithm) requires protected group memberships at the input to generate cluster outputs, we cannot directly use it for attacks that are designed for traditional clustering models and aim to reduce their clustering performance as these protected group memberships do not exist. Also, note that providing dummy protected group memberships might lead to solutions that are being affected by the choice of these group memberships and not necessarily the attack itself.
>
> To accommodate Reviewer EUnH's suggestion, we can forego the second stage of CFC (which utilizes fairness) and just employ the first stage to generate cluster outputs using the obtained co-association matrix and single-linkage clustering as in previous work [1]. For this modified approach, we can then carry out experiments using two recent attacks designed for clustering models: the poisoning attack by Cina et. al, [2] and a new inference attack against deep learning based clustering [3]. Both these attacks seek to decrease the clustering performance. For consistency between the original attack implementations, here we use the _Fashion-MNIST_ and _CIFAR-10_ datasets (sampled and prepared as in [2]), and poison 15\% attack samples using both approaches. We use these datasets for analysis since they are utilized in the original attack paper [2]. We then measure clustering performance for both these approaches using the NMI before the attack and after the attack. These results are provided as a table below. As can be seen, while the modified CFC model is not completely unaffected by the attack, performance is not deteriorated fully either. While this is by no means an exhaustive set of experiments to conclude robustness, for future work this can be explored in more detail along the same direction.
>
> | Attack |    Dataset    |  Pre-Attack (NMI) | Post-Attack (NMI) | Change (\%) |
> |:------:|:-------------:|:-----------------:|:-----------------:|:-----------:|
> |   [2]  |    _CIFAR-10_   | $0.901 \pm 0.009$ | $0.743 \pm 0.005$ |  $(-)17.53$ |
> |   [2]  | _Fashion-MNIST_ | $0.927 \pm 0.001$ | $0.602 \pm 0.018$ |  $(-)35.05$ |
> |   [3]  |    _CIFAR-10_   | $0.931 \pm 0.000$ | $0.893 \pm 0.003$ |  $(-)4.081$ |
> |   [3]  | _Fashion-MNIST_ | $0.926 \pm 0.002$ | $0.668 \pm 0.004$ |  $(-)27.86$ |
>
>
> [1] Fred, Ana LN, and Anil K. Jain. "Combining multiple clusterings using evidence accumulation." IEEE transactions on pattern analysis and machine intelligence 27.6 (2005): 835-850.
>
> [2] Cinà, Antonio Emanuele, Alessandro Torcinovich, and Marcello Pelillo. "A black-box adversarial attack for poisoning clustering." Pattern Recognition 122 (2022): 108306.
>
> [3]    Chhabra, Anshuman, Ashwin Sekhari, and Prasant Mohapatra. "On the Robustness of Deep Clustering Models: Adversarial Attacks and Defenses." arXiv preprint arXiv:2210.01940 (2022).

---

### Official Review · Reviewer_HmWf · 2022-10-24

**Confidence:** 4
**Correctness:** 4
**Technical Novelty And Significance:** 4
**Empirical Novelty And Significance:** 3
**Recommendation:** 8

**Clarity, Quality, Novelty And Reproducibility:**

### Clarity

The paper is well written and quite clear, except for a few details:

- Page 8, right before equation $(3)$, the “cluster $c_k$” are mentioned but I still did not understand where they came from.
- Page 8, between equation $(3)$ and $(4)$, the meaning of $P(g)$ seems unclear to me. Why would $P$ take $g$ as an argument when the $P$ has allegedly been constructed with the probabilities $p_k^x$ which do not seem to depend on $g$.
- Page 13, in Balance definition, the symbol $r_k$ seem confusing. Balance has been generalized to more than two groups in Suman K. Bera’s *Fair Algorithms for Clustering* and a similar notation explicitly showing the group considered such as $r_k^g$ would have been preferred.

### Quality

One of the claim in the paper seems confusing. At page 7, the authors write: “*Since attacked samples are a tiny portion of the whole training data, the probability of these being selected into the subset is also small, which decreases their negative impact.*” But since all of the samples are used in a basic partition (since $\cup_{i=1}^r X_i = X$), the reason for the robustness of CFC must be attributed to another mechanism. This question is later investigated in appendix G but further work would be required (comparing several values of $r$?)

### Novelty

This work is undoubtedly novel as the threat model and downstream task do not seem to appear in other works.

### Reproducibility

Reproducilbility cannot be evaluated as such without the source code but the given implementation details are highly appreciated. Solutions exist to ensure confidentiality or uploaded in supplementary material.

**Details Of Ethics Concerns:**

**This paper exhibit an attack to which state-of-the-art algorithms are vulnerable but proposes a defense to that attack.** It is of high importance for users of the vulnerable algorithms to be aware of their flaws and the existing solutions. Works proposing both an attack and a defense are rare enough for me to emphasise on how necessary it is.

**Strength And Weaknesses:**

### Strengths

- Being *black-box* and exploring a wide range of ratio of poisoned data, the threat model seems highly relevant;
- Brings neural network in the field of fair machine learning, which seems to still be understudied (as showed by the state-of-the-art algorithms used to compare to in the paper);
- The paper is well written and easy to read.

### Weaknesses

- Some comparisons would have benefited being more thorough:
    - Figure 2 and 4 would have benefited showing results on CFC;
    - Table 2 and 4 would have benefited comparing with random attacks;
    - Figure 6 experiments would have benefited from a confidence interval;
    - Comparing different fair algorithms that could be used to generate the $J$ parameter in equation $(4)$.
- Even though an in-depth exploration of CFC is proposed in appendix, an ablation study would have been relevant, especially on the different losses being used.
- Studying the robustness of CFC against other known attacks on clustering (relaxing some of the problem constraints if necessary) would have been highly appreciated.
- Is the ratio of poisoned samples reasonable? 15% seems like a lot, it would be interesting to see how CFC compare with a low number of adversarial samples (1%?)

**Summary Of The Paper:**

This paper tackles **unsupervised fair clustering**. The authors first propose **a black-box adversarial attack** on state-of-the-art fair clustering algorithms, effective on a toy dataset they propose as well as real datasets (MNIST, USPS, Office-31). In their threat model, an adversary can modify *the protected attribute* of a subset of the dataset. Its goal is to minimize a fairness utility function while trying not to degrade the performances. Success of the attack is evaluated on the remaining untouched datapoints.
In a second time, the authors propose **a defense against that attack**, an unsupervised fair clustering algorithm which is robust to the attack.

**Summary Of The Review:**

The novelty and relevance of this work make it interesting for the scientific community in both Robust Machine Learning and Fairness-related Machine Learning research. Although I pointed a few (alleged) weaknesses, the paper should have tackled, they nonetheless exhibited a relevant attack and experimentally demonstrated a defense to that attack. Further work on this topic will be of great value for the community.

---

> ### Author Response · Authors · 2022-11-16
> **Response to Reviewer HmWf [1/3]**
>
> We thank Reviewer HmWf for their detailed analysis of our paper and for the constructive feedback provided. We have aimed to alleviate some of the concerns raised by providing additional experiments or detailed justifications for our decisions.
>
> **Figure 2 \& 4**. We agree with Reviewer HmWf on showing CFC trends in the attack figures. However, as Figure 2 precedes Section 3 this is somewhat complicated to do, since the CFC model has not been introduced yet in the paper. Since this issue does not exist with Figure 4, we have now updated it to also include CFC trends. If the reviewer has some suggestions for how they would like us to incorporate CFC into Figure 2, we can also do that (however note that Figure 6 presents the same results but as pre-attack/post-attack ratios). Please advise.
>
> **Table 2 \& 4**. We agree with Reviewer HmWf and have run experiments using the random attack and due to size/space constraints we have provided updated versions of Table 2 and Table 4 as **Table 6** and **Table 7** in **Appendix I.2**, respectively. The random attack is also unable to reduce the efficacy and utility of CFC significantly, and considerably less so than our proposed attack.
>
>
> **Figure 6**. We have updated Figure 6 to include confidence intervals.
>
> **Different fair algorithms in Eq. (4)**.  According to earlier experimental analysis, we found that the choice of fair clustering algorithm does not largely affect the final clustering results since the goal is to just preserve structure and guide the learning model to not opt for trivial degenerate solutions. To demonstrate this we present results in **Table 8, Appendix I.3** on *Office-31* for CFC with the other fair clustering algorithms (KFC and FSC) being used to generate $J$. As can be seen, performance is generally similar pre- and post-attack irrespective of how $J$ was generated.
>
> **Ablation study**. We thank Reviewer HmWf for this suggestion. Actually, we have also thought about the ablation study before. Unfortunately, an ablation study on the losses poses some issues which is why we opted for the in-depth exploration instead. Our primary aim is to measure performance of the model before and after the attack. For an ablation study, removing the clustering loss would not give us any clusters at the output head which is why we would be unable to attack. Similarly, if we remove the structural preservation loss we just end up with the degenerate solutions discussed in prior work [1], again preventing us from carrying out the attack.
>
> **Other known attacks on CFC**. We thank Reviewer HmWf for this suggestion as well. Since CFC (and any other fair clustering algorithm) requires protected group memberships at the input to generate cluster outputs, we cannot directly use it for attacks that are designed for traditional clustering models and aim to reduce their clustering performance as these protected group memberships do not exist. Also, note that providing dummy protected group memberships might lead to solutions that are being affected by the choice of these group memberships and not necessarily the attack itself.
>
> To accommodate Reviewer HmWf's suggestion, we can forego the second stage of CFC (which utilizes fairness) and just employ the first stage to generate cluster outputs using the obtained co-association matrix and single-linkage clustering as in previous work [2]. For this modified approach, we can then carry out experiments using two recent attacks designed for clustering models: the poisoning attack by Cina et. al, [3] and a new inference attack against deep learning based clustering [4]. Both these attacks seek to decrease the clustering performance. For consistency between the original attack implementations, here we use the _Fashion-MNIST_ and _CIFAR-10_ datasets (sampled and prepared as in [3]), and poison 15\% attack samples using both approaches. We use these datasets for analysis since they are utilized in the original attack paper [3]. We then measure clustering performance for both these approaches using the NMI before the attack and after the attack. These results are provided below as a table. As can be seen, while the modified CFC model is not completely unaffected by the attack, performance is not deteriorated fully either. While this is by no means an exhaustive set of experiments to conclude robustness, for future work this can be explored in more detail along the same direction.
>
> | Attack |    Dataset    |  Pre-Attack (NMI) | Post-Attack (NMI) | Change (\%) |
> |:------:|:-------------:|:-----------------:|:-----------------:|:-----------:|
> |   [3]  |    _CIFAR-10_   | $0.901 \pm 0.009$ | $0.743 \pm 0.005$ |  $(-)17.53$ |
> |   [3]  | _Fashion-MNIST_ | $0.927 \pm 0.001$ | $0.602 \pm 0.018$ |  $(-)35.05$ |
> |   [4]  |    _CIFAR-10_   | $0.931 \pm 0.000$ | $0.893 \pm 0.003$ |  $(-)4.081$ |
> |   [4]  | _Fashion-MNIST_ | $0.926 \pm 0.002$ | $0.668 \pm 0.004$ |  $(-)27.86$ |

---

> > ### Author Response · Authors · 2022-11-16
> > **Response to Reviewer HmWf [2/3]**
> >
> > **The size of $G_A$**. We thank the reviewer for this comment. We answer this in three ways:
> >
> > [_Consumption_]. Even though in the tabular results the attacker can control 15\% of the protected group memberships, in actuality, this is an upper-bound in the attack optimization and the attacker might not necessarily "consume" all these samples by poisoning them. We believe that actual consumption is much lower than 15\%. In particular, we conducted a new experiment to analyze this for the 15\% case. These results are provided below as a table. As can be seen in the table, the "consumption" is actually much lower than 15\% and closer to 7\% for all three fair algorithms on all datasets.
> >
> > | Algorithm | _Office-31_ | _MNIST-USPS_ | _DIGITS_ | _Yale_ |
> > |-----------|-----------|------------|--------|-------|
> > | SFD       | 7.781     | 7.434      | 7.625  | 7.545 |
> > | FSC       | 7.476     | 7.468      | 7.377  | 7.384 |
> > | KFC       | 7.359     | 7.378      | 7.385  | 7.520 |
> >
> >
> > [_Results for the 3.75\% case_]. In **Table 5** in **Appendix I.1**, we have now also provided results for the 3.75\% case, similar to Table 1 where 15\% group membership labels were switched. Here it can be seen that the attack is very well-performing despite the small \% of samples that are allowed to be perturbed.
> >
> > [_Trends of Figure 2_]. In Figure 2 we have shown the changing values of the fairness and performance metrics measured on $G_D$ (the benign samples) with increasing \% of protected group memberships (poisoned samples) that the attacker can control on the x-axis. Here it can be seen that our attack fairly quickly reduces fairness utility, starting around 3.75\% for both the _MNIST-USPS_ and _Office-31_ datasets.
> >
> > **Clarifications**. We apologize for the lack of clarity. Here we provide more explanations.
> >
> > [$c_k$]. This is actually a typo and should read "cluster centroid $c_k$" instead of "cluster $c_k$." The embedding space centroids are initially randomly initialized as in previous approaches [5] and update as the gradient backpropagates and refines the embedding space. The loss used in our paper also incorporates protected group memberships for fairness regularization similar to [1].
> >
> > [$P(g)$]. Here $P$ is not taking $g$ as an argument but instead we are only looking at the subset of samples in $P$ that belong to protected group $g$. We have since updated the notation here to read as $P_g$ and improved the writing as well.
> >
> > [$r_k$]. We have updated the definition of Balance to more closely reflect the notation used in the original definition of Bera et al's paper [6].
> >
> > **Statement on Page 7**. We would like to expand on this statement. Since each partition itself is a small subset of the entire dataset, it will be probabilistically low for the attack samples to be selected in each partition. Next, K-means is undertaken on each of these partitions (no group membership information is used), and we will obtain $r$ clusterings for each of the $r$ partitions. Now, even though the attack samples are selected in some of the partitions, the overall consensus matrix generated as part of the first stage acts as a clear guide for how these points should be eventually clustered. As this matrix is provided as input to the second stage of CFC, we believe the attack samples' poisoned group memberships themselves are not "good enough" to influence the clustering model to reduce fairness for samples in $G_D$.

---

> > > ### Author Response · Authors · 2022-11-16
> > > **Response to Reviewer HmWf [3/3]**
> > >
> > > **Flaws of the existing solutions.** We thank Reviewer HmWf for the suggestion. We have now added a discussion on ethics as a separate section in the main paper (**Section 6**) as follows:
> > >
> > > *In this paper, we have proposed a novel adversarial attack that aims to reduce the fairness utility of fair clustering models. Furthermore, we also propose a defense model based on consensus clustering and fair graph representation learning that is robust to the aforementioned attack. Both the proposed attack and the defense are important contributions that help facilitate ethics in ML research, due to a number of key reasons. First, there are very few studies that investigate the influence of adversaries on the fairness of unsupervised models (such as clustering), and hence, our work paves the way for future work that can study the effect of such attacks in other different learning models. Understanding the security vulnerabilities of models, especially with regard to fairness is the first step to developing models that are more robust to such attacks. Second, even though our attack approach can have a negative societal impact in the wrong hands, we propose a defense approach that can be used as a deterrent against such attacks. Third, we believe our defense approach can serve as a starting point for the development of more robust and fair ML models. Through this work, we seek to underscore the need for making fair clustering models robust to adversarial influence, and hence, drive the development of truly fair robust clustering models.*
> > >
> > > **Reproducibility**. We sincerely apologize for not including the code; it was an oversight on our part. We have now added the code here: <https://anonymous.4open.science/r/CFC>.
> > >
> > > [1] Li, Peizhao, Han Zhao, and Hongfu Liu. "Deep fair clustering for visual learning." Proceedings of the IEEE/CVF Conference on Computer Vision and Pattern Recognition. 2020.
> > >
> > > [2] Fred, Ana LN, and Anil K. Jain. "Combining multiple clusterings using evidence accumulation." IEEE transactions on pattern analysis and machine intelligence 27.6 (2005): 835-850.
> > >
> > > [3] Cinà, Antonio Emanuele, Alessandro Torcinovich, and Marcello Pelillo. "A black-box adversarial attack for poisoning clustering." Pattern Recognition 122 (2022): 108306.
> > >
> > > [4] Chhabra, Anshuman, Ashwin Sekhari, and Prasant Mohapatra. "On the Robustness of Deep Clustering Models: Adversarial Attacks and Defenses." arXiv preprint arXiv:2210.01940 (2022).
> > >
> > > [5] Xie, Junyuan, Ross Girshick, and Ali Farhadi. "Unsupervised deep embedding for clustering analysis." International conference on machine learning. PMLR, 2016.
> > >
> > > [6] Bera, Suman, et al. "Fair algorithms for clustering." Advances in Neural Information Processing Systems 32 (2019).

---

### Official Review · Reviewer_XY3W · 2022-10-25

**Confidence:** 2
**Correctness:** 3
**Technical Novelty And Significance:** 2
**Empirical Novelty And Significance:** 3
**Recommendation:** 8

**Clarity, Quality, Novelty And Reproducibility:**

The paper is mainly clear (except for section 3.1). The ideas are somewhat novel as similar attacks have been studied for the clustering objective (with no fairness considerations). I do not see any link to the code but most of the experimental details are described somewhat clearly.

**Strength And Weaknesses:**

------------------------
Strengths:
------------------------
-- The paper is mainly well written (except section 3.1) and easy to follow.

-- The idea of adversarial attacks on fair clustering is natural and worth exploring.

------------------------
Weaknesses:
------------------------
-- The paper would be strengthened if the authors provide a clear motivating example.

-- The size of $G_A$ in all the experiments is rather large. In all the definitions, it is assumed that this proportion is small but all of the experiments focus on 15% which is rather large. How do the results change for proportions much smaller than 15%?

------------------------
Minor Comments:
------------------------
-- What is the axis in Figure 2? It would be nice to have labels for graphs.

-- Can you elaborate on why the fairness violation can decrease after accounting for attacks? This intuitively does not make sense as the approach which only optimizes for the fairness violation should achieve a smaller loss.


**Summary Of The Paper:**

The paper studies adversarial attacks on fair clustering. The authors demonstrate simple adversarial attacks on existing fair clustering approaches and propose an algorithm that is robust to the attack.

**Summary Of The Review:**

Overall, I think this is interesting work, though I am not an expert in the area.

-------------------------
Post Rebuttal:
-------------------------
I thank the authors for their response. After reading all the reviews and responses, I have increased my score from borderline accept to accept.

---

> ### Author Response · Authors · 2022-11-16
> **Response to Reviewer XY3W**
>
> We thank Reviewer XY3W for their thorough reading of our work, and for their detailed feedback. We have responded to concerns below:
>
> **Motivating example.** We have provided one example scenario in the manuscript in Section 2.1. However, based on Reviewer XY3W's feedback, here we provide another more detailed motivating example based on societal resource allocation (e.g., bank loan disbursement). We have added this to the manuscript as well. The example is as follows:
>
> _Consider a societal resource allocation problem, for example, job shortlisting or loan approval. Given that legal regulations should exist in the near future that require the use of fair algorithms in such applications [1], a fair clustering algorithm can be utilized in the back-end, assuming the protected group to be sex and female is the minority group. For anarchistic or profit related purposes, by controlling a small percentage of samples' protected group memberships (either by gaining access to the system, social engineering, etc.), the attacker can then aim to reduce the fairness of the model (balance) for any samples not under their control and lead to unfair outputs that discriminate against the minority group (i.e., the samples belong to the female group in our case). Thus, using our attack the attacker completely invalidates the fairness provided by the fair algorithm._
>
> **The size of $G_A$**. We thank the reviewer for this comment. We answer this in three ways:
>
> [_Consumption_]. Even though in the tabular results the attacker can control 15\% of the protected group memberships, in actuality, this is an upper-bound in the attack optimization and the attacker might not necessarily "consume" all these samples by poisoning them. We believe that actual consumption is much lower than 15\%. In particular, we conducted a new experiment to analyze this for the 15\% case. These results are provided as a table below. As can be seen in the table, the "consumption" is actually much lower than 15\% and closer to 7\% for all three fair algorithms on all datasets.
>
> | Algorithm | _Office-31_ | _MNIST-USPS_ | _DIGITS_ | _Yale_ |
> |-----------|-----------|------------|--------|-------|
> | SFD       | 7.781     | 7.434      | 7.625  | 7.545 |
> | FSC       | 7.476     | 7.468      | 7.377  | 7.384 |
> | KFC       | 7.359     | 7.378      | 7.385  | 7.520 |
>
>
>
> [_Results for the 3.75\% case_]. In **Table 5** in **Appendix I.1**, we have now also provided results for the 3.75\% case , similar to Table 1 where 15\% group membership labels were switched. Here it can be seen that the attack is very well-performing despite the small \% of samples that are allowed to be perturbed.
>
>
>
> [_Trends of Figure 2_]. In Figure 2 we have shown the changing values of the fairness and performance metrics measured on $G_D$ (the benign samples) with increasing \% of protected group memberships (poisoned samples) that the attacker can control on the x-axis. Here it can be seen that our attack fairly quickly reduces fairness utility, starting around 3.75\% for both the _MNIST-USPS_ and _Office-31_ datasets.
>
>
> **Axis in Figure 2**. We apologize for this oversight on our part. The axis represents the \% of group memberships the attacker can control as $G_A$. Hence, the figure shows how the metrics vary with more poisoning carried out by the attacker. We have updated the caption to reflect the axis label.
>
> **Fairness violation decreases**. The attack optimization problem essentially consists of poisoning samples in $G_A$ such that fairness utility for the benign samples in $G_D$ decreases. Thus, the attacker can optimize $G_A$ in such a way that the original model will try to obtain fair solutions for the whole dataset, but in doing so fairness violation for points in $G_D$ will increase in comparison to the original values prior to the attack. Thus, the attacker can leverage the poisoned sample set $G_A$ to drive the fair model to converge to solutions that might still be somewhat fair on the combined data including $G_A$, but become unfair to samples in $G_D$.
>
> **Section 3.1**. Sorry for the bad reading experience. We sincerely solicit Reviewer XY3W to provide more details and suggestions. We are happy to address them and improve readability and understanding.
>
>
> **Code link**. We sincerely apologize for not including the code; it was an oversight on our part. We have now added the code here: <https://anonymous.4open.science/r/CFC>.
>
>
> [1] Petrasic, Kevin, et al. "Algorithms and bias: What lenders need to know." White \& Case (2017).

---

### Official Review · Reviewer_UuYT · 2022-10-25

**Confidence:** 4
**Correctness:** 4
**Technical Novelty And Significance:** 3
**Empirical Novelty And Significance:** 3
**Recommendation:** 6

**Clarity, Quality, Novelty And Reproducibility:**

Clarity: The writing is good and clear. The empirical results can verify the proposed conclusions. A comment is that the authors should provide an ethnic discussion since the topic is about fairness.
Quality: The proposed black-box attack is interesting, but lacks some theoretical analysis. For instance, what is the worst influence of such a black-box attack on some known fair clustering algorithms?
Novelty: The considered problem is novel.

**Strength And Weaknesses:**

Strength:
- The proposed problem of robust fair clustering is interesting and novel.
- The proposed black-box attack is natural in robust clustering literature.
- The empirical results that existing fair clustering algorithms are highly susceptible to adversarial influence are convincing.

Weaknesses:
- The proposed approach CFC does not have any provable guarantees, both in fairness and in accuracy. I expect to see at least some performance analysis for CFC.

**Summary Of The Paper:**

The paper studies the fairness attack and defense problem. The authors propose a black-box attack against fair clustering algorithms that works by perturbing a small percentage of samples’ protected group memberships. They also proposed a defense algorithm, named Consensus Fair Clustering (CFC), that utilizes consensus clustering along with fairness constraints to output robust and fair clusters. Empirically, existing fair clustering algorithms are highly susceptible to adversarial influence, while the proposed CFC algorithm is highly effective and robust as it resists the proposed fairness attack well.

**Summary Of The Review:**

The paper studies the fairness attack and defense problem and proposes a novel black-box attack. The empirical results that existing fair clustering algorithms are highly susceptible to adversarial influence are convincing. The major concern is the lack of theoretical analysis on both the influence of black-box attacks and the proposed CFC approach.

Overall, I tend to accept the paper. I think the paper can be further improved by providing some theoretical analysis.

---

> ### Author Response · Authors · 2022-11-16
> **Response to Reviewer UuYT**
>
> We would like to thank Reviewer UuYT for their insightful comments and suggestions. We have responded to the concerns below:
>
> **Provable guarantees**. Thank you for this great suggestion. In the beginning, we planned to provide some theoretical analysis on both fair attack and defense. And we have demonstrated our fairness attack is effective against k-center/k-median fairlet decomposition based clustering algorithms (please refer to the third point below, worst influence). Later, due to the time and page limit, we have figured out it was difficult to add them into the current manuscript.
>
> A provable guarantee makes our CFC more stronger and trustworthy. We would like to analyze the robustness and generalization of CFC, followed by the robustness property of consensus clustering [1] and known results on Graph-MLP [2]. Due to complex and highly non-convex losses in CFC, it is non-trivial to undertake theoretical analysis during the short response period. Firmly, we will dive more along this new and promising direction in the future. We believe robust fair clustering with theoretical guarantees will vastly improve our understanding of how to prevent fairness attacks in a more informed manner, and lead to the development of more resilient clustering algorithms.
>
> **Ethics discussion**. We thank Reviewer UuYT for the suggestion. We have now added a discussion on ethics as a separate section in the main paper (**Section 6**) as follows:
>
> *In this paper, we have proposed a novel adversarial attack that aims to reduce the fairness utility of fair clustering models. Furthermore, we also propose a defense model based on consensus clustering and fair graph representation learning that is robust to the aforementioned attack. Both the proposed attack and the defense are important contributions that help facilitate ethics in ML research, due to a number of key reasons. First, there are very few studies that investigate the influence of adversaries on the fairness of unsupervised models (such as clustering), and hence, our work paves the way for future work that can study the effect of such attacks in other different learning models. Understanding the security vulnerabilities of models, especially with regard to fairness is the first step to developing models that are more robust to such attacks. Second, even though our attack approach can have a negative societal impact in the wrong hands, we propose a defense approach that can be used as a deterrent against such attacks. Third, we believe our defense approach can serve as a starting point for the development of more robust and fair ML models. Through this work, we seek to underscore the need for making fair clustering models robust to adversarial influence, and hence, drive the development of truly fair robust clustering models.*
>
> **Worst influence**. We appreciate the suggestion. To analyze fairness attacks from a theoretical lens, usually some assumption is made on the data distribution, such as in [3]. Thus, we utilize the assumption of well-separated clusters as in [3] to provide simple analysis of our attack's efficacy against the fairlet decomposition fair clustering approach of [4] which the SFD fair clustering algorithm considered in our paper is based on. This new theoretical analysis is provided in **Appendix D**.
>
> [1] Liu, Hongfu, et al. "Spectral ensemble clustering via weighted k-means: Theoretical and practical evidence." IEEE transactions on knowledge and data engineering 29.5 (2017): 1129-1143.
>
> [2] Wu, Lirong, and Stan Z. Li. "Beyond Message Passing Paradigm: Training Graph Data with Consistency Constraints." (2021).
>
> [3] Chhabra, Anshuman, Adish Singla, and Prasant Mohapatra. "Fairness Degrading Adversarial Attacks Against Clustering Algorithms." arXiv preprint arXiv:2110.12020 (2021).
>
> [4] Chierichetti, Flavio, et al. "Fair clustering through fairlets." Advances in Neural Information Processing Systems 30 (2017).

---

### Decision · Program_Chairs · 2023-01-20

**Decision:**

Accept: poster

**Justification For Why Not Higher Score:**

The results is nice and interesting but not strong enough to be a spotlight

**Justification For Why Not Lower Score:**

The contribution is novel and study a well-know problem from a different and interesting perspective.

**Metareview: Summary, Strengths And Weaknesses:**

The authors study the classic fair classic problem. They first propose a novel black-box attack and show that a very simple strategy can destroy the guarantees of previous algorithm. Then they present a new technique and show experimentally that it provides better guarantees that the previous result.

The paper is well-written and it study a classic problem from a novel perspective so it would be a nice addition to the ICLR program.

**Note From Pc:**

if the above contains the word "oral" or "spotlight" please see: "oral" presentation means -> notable-top-5% and "spotlight" means -> notable-top-25%. As stated in our emails, we are disassociating presentation type from AC recommendations